# CALIPSO Level 3 Stratospheric Aerosol Profile Product: Version 1.00 Algorithm Description and Initial Assessment

Jayanta Kar[1,2], Kam-Pui Lee[1,2], Mark A. Vaughan[2], Jason L. Tackett[2], Charles R. Trepte[2], David M. Winker[2], Patricia L. Lucker[1,2], Brian J. Getzewich[1,2]

[1]Science Systems and Applications Inc., Hampton, VA, USA
[2]NASA Langley Research Center, Hampton, VA, USA

*Correspondence to*: J. Kar (jayanta.kar@nasa.gov)

**Abstract.** In August 2018, the Cloud-Aerosol Lidar and Infrared Pathfinder Satellite Observation (CALIPSO) project released a new level 3 stratospheric aerosol profile data product derived from nearly 12 years of measurements acquired by the space-borne Cloud-Aerosol Lidar with Orthogonal Polarization (CALIOP). This monthly averaged, gridded level 3 product is based on version 4 of the CALIOP level 1B and level 2 data products, which feature significantly improved calibration that now makes it possible to reliably retrieve profiles of stratospheric aerosol extinction and backscatter coefficients at 532 nm. This paper describes the science algorithm and data handling techniques that were developed to generate the CALIPSO version 1.00 level 3 stratospheric aerosol profile product. Further, we show that the extinction profiles (retrieved using a constant lidar ratio of 50 sr) capture the major stratospheric perturbations in both hemispheres over the last decade resulting from volcanic eruptions, extreme smoke events, and signatures of stratospheric dynamics. Initial assessment of the product by inter-comparison with the stratospheric aerosol retrievals from the Stratospheric Aerosol and Gas Experiment III (SAGE III) on the International Space Station (ISS) indicates good agreement in the tropical stratospheric aerosol layer (30°N-30°S), where the average difference between zonal mean extinction profiles is typically less than 25% between 20 km and 30 km (CALIPSO biased high). However, differences can exceed 100% in the very low aerosol loading regimes found above 25 km at higher latitudes. Similarly there are large differences (≥100%) within 2 to 3 kilometers above the tropopause which might be due to cloud contamination issues.

## 1. Introduction.

While the bulk of the global distribution of atmospheric aerosols is concentrated within the planetary boundary layer and free troposphere, the persistent aerosol burden in the stratosphere has long been known to have important implications for Earth's climate (Turco et al., 1980). Techniques for reliable detection of a background aerosol layer in the stratosphere date back to the early 1960s (Junge and Manson, 1961). These aerosols are mostly liquid sulfate particles which are derived from precursor gases like $SO_2$ and carbonyl sulfide (OCS) transported from the troposphere (Thomason and Peter, 2006, Kremser et al., 2016, Thomason et al., 2018). In addition, intermittent volcanic eruptions and strong biomass burning events can inject sulfates, ash, and smoke into the stratosphere, which can last for long periods of time and exert significant climatic influences. For example, stratospheric perturbations from the Pinatubo volcano in 1991 lasted for several years (Chazette et al., 1995, Robock, 2000, Deshler, 2008). While eruptions of the same scale as Pinatubo have not taken place in the last 25 years or so, there is evidence that a large number of smaller eruptions has been significantly affecting the stratosphere with implications for the climate system (Vernier et al., 2011a, Solomon et al., 2011). Thus it is very important to monitor the stratospheric aerosol loading over the long term. In pursuit of this goal, stratospheric aerosol measurements have been made using numerous techniques, including ground based lidars and balloon borne in situ samplers, as well as multi-sensor aircraft measurements, since the mid-twentieth century (Junge et al., 1961, Northam et al., 1974, Hoffman et al., 1975, McCormick et al., 1984, Gramms and Fiocco, 1986, Brock et al., 1993, Beyerle et al., 1994, Jaeger and Deshler 2002).

Most of our current knowledge of the global distribution of stratospheric aerosols comes from satellite measurements. The earliest such measurements were carried out by the Stratospheric Aerosol Measurement II (SAM II) on board the Nimbus 7 spacecraft which provided the vertical profiles of aerosol extinction at 1 μm and were followed by the Stratospheric Aerosol and Gas Experiment (SAGE) series of instruments (Chu and McCormick, 1979, Kent and McCormick, 1984, Mauldin et al., 1985; Chu et al., 1993; Damadeo et al., 2013). The basic principle employed in these instruments is solar occultation, wherein the vertical profile of stratospheric aerosols is retrieved from measurement of sunlight as the rays pass through the atmosphere during sunrise and sunset events as observed from the orbiting spacecraft. Stratospheric aerosols have been characterized using this technique from SAGE instruments on Earth Radiation Budget Satellite

(ERBS) and Meteor-3M as well as from the International Space Station (ISS). Among other space-borne instruments that have used this technique are the Polar Ozone and Aerosol Measurement (POAM II, POAM III, Glaccum et al., 1996, Lucke et al., 1999) and Measurement of Aerosol Extinction in the Stratosphere and Troposphere Retrieved by Occultation (MAESTRO, McElroy et al., 2009). In addition, the Optical Spectrograph and InfraRed Imager System (OSIRIS) and the Ozone Mapping and Profiler Suite (OMPS) have used a limb scatter technique to obtain the aerosol extinction profiles (Bourassa et al., 2012, Chen et al., 2018).

A novel and pioneering technique to retrieve aerosol profiles from space came about with the launch of the Cloud-Aerosol Lidar and Infrared Pathfinder Satellite Observation (CALIPSO) mission in April 2006, with a two-wavelength, polarization-sensitive elastic backscatter lidar as the primary payload (Winker et al., 2010). For over 12 years the Cloud-Aerosol Lidar with Orthogonal Polarization (CALIOP) has been providing vertically-resolved profiles of aerosol and cloud extinction globally. The primary measurement from a space-borne elastic backscatter lidar consists of the attenuated backscatter coefficients of the aerosols and clouds in the atmosphere. The strong backscatter from the tropospheric aerosols, combined with CALIOP's relatively strong signal-to-noise ratio (SNR), has been exploited to provide accurate extinction profiles in the troposphere (Young and Vaughan, 2009, Winker et al., 2013, Young et al., 2013, 2016, 2018). In comparison, the aerosol loading in the stratosphere is much lower with correspondingly smaller SNR. As such, retrieving stratospheric aerosol information was not originally a principal target of the CALIPSO mission. However, early results indicated that it might be possible to obtain such information with sufficient averaging of the data (Thomason and Pitts, 2007, Vernier et al., 2009).

One of the issues impacting the retrieval of stratospheric aerosol extinction was the realization that the standard calibration altitude of CALIOP, which was originally fixed at 30-34 km (Powell et al., 2009), was not completely free of aerosols and thus applying the molecular normalization technique at these altitudes would bias the aerosol extinction profiles (Vernier et al., 2009). This issue has since been addressed with the release of the version 4 (V4) family of CALIPSO data products in November 2016. In this version, the calibration altitude for the nighttime 532 nm data which is the primary calibration for all of CALIOP measurements (all the other measurements like the daytime data as well as the 1064 nm data are calibrated relative to the 532 nm nighttime calibration) was raised to 36-39 km, where the aerosol loading is expected to be negligible (Kar et al., 2018a). This largely removed the aerosol contamination issue, making

reliable retrievals of stratospheric aerosols possible. Accordingly, a stand-alone CALIPSO stratospheric aerosol profile product was developed which uses the V4 level 1B and level 2 data from the CALIOP measurements. This is a level 3 monthly averaged product gridded in latitude (5°), longitude (20°) and altitude (900 m). In what follows, we describe the overall algorithm and its implementation in detail in section 2. Section 3 then presents a comprehensive assessment of the quality and capabilities of this new data product, including analyses of the temporal and spatial evolution of specific stratospheric features captured by the product and inter-comparisons with extinction retrievals from SAGE III on ISS. Discussion and concluding remarks are given in section 4 and section 5 respectively.

## 2. Overall design of the level 3 stratospheric aerosol profile product

### 2.1 Motivation for a CALIPSO stratospheric product

The CALIPSO level 3 stratospheric aerosol profile product is built primarily from the V4 level 1B 532 nm attenuated backscatter profiles (https://eosweb.larc.nasa.gov/project/calipso/cal_lid_l1-standard-v4-10) As mentioned above, the most fundamental change in V4 level 1 data was the improved calibration of the 532 nm nighttime data (Kar et al., 2018a). The consequences of this change are illustrated in Figure 1, which shows the median values of zonally averaged attenuated scattering ratios at 30-34 km from version 3 (V3) and V4 for the month of May 2009.

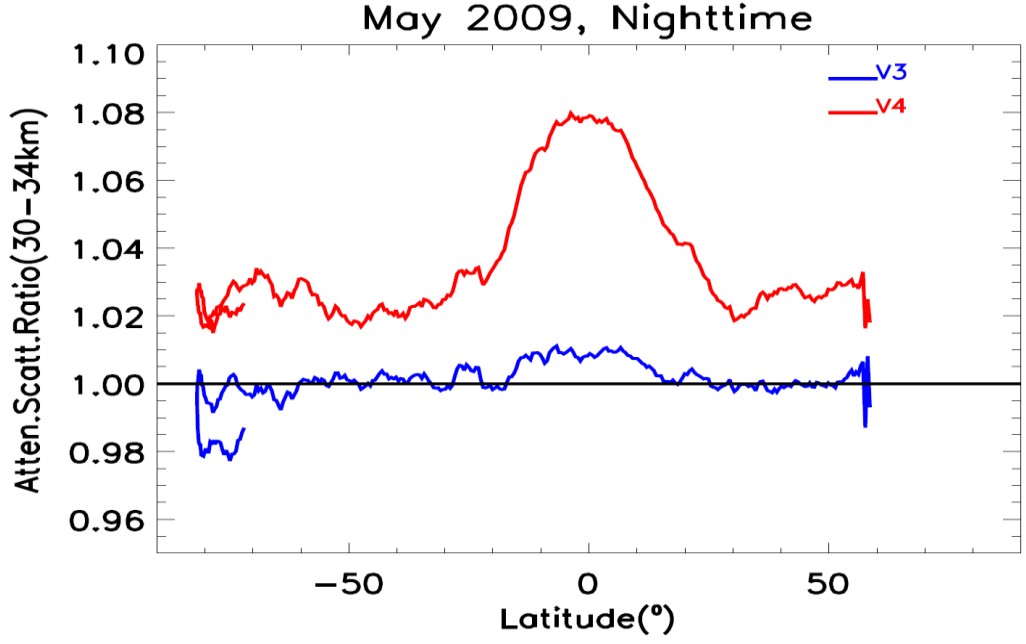

**Figure 1.** Median values of zonally and vertically (over 30-34 km) averaged 532 nm attenuated scattering ratios for May 2009 nighttime data from V3 and V4. Data over the South Atlantic Anomaly were excluded. A 10-point smoothing of the data has been applied.

As shown in Eq. (1), the attenuated scattering ratios, $R'(z)$, are computed as the ratio of the measured attenuated backscatter coefficients, $\beta'_{measured}(z)$, which contain contributions from both molecular and particulate backscatter ($\beta_m(z)$ and $\beta_p(z)$, respectively), and the attenuated backscatter coefficients calculated from modeled profiles of molecular number densities, $\beta'_{modeled}(z)$ (Vaughan et al., 2009).

$$R'(z) = \frac{\beta'_{measured}(z)}{\beta'_{modeled}(z)} = \frac{\left(\beta_m(z) + \beta_p(z)\right) T_m^2(z) T_{O_3}^2(z) T_p^2(z)}{\beta_{m,modeled}(z) T_{m,modeled}^2(z) T_{O_3,modeled}^2(z)} = \left(1 + \frac{\beta_p(z)}{\beta_{m,modeled}(z)}\right) T_p^2(z) \tag{1}$$

In this expression, $T_X^2(z)$ represents the two-way transmittance (i.e., signal attenuation) between the lidar and altitude z for air molecules (X = m), ozone (X = $O_3$), and particulates (X = p). In V3, the calibration region was fixed at 30-34 km, with the assumption that the aerosol loading in this region was negligible (Powell et al., 2009); i.e., $\beta_p(z) \approx 0$ and $T_p^2(z) = 1$. This assumption essentially forced the V3 attenuated scattering ratios in the region to one. For the V4 data release, the calibration region was raised to 36-39 km, with the concomitant assumption that the mean scattering ratio at these higher altitudes is $1.01 \pm 0.01$. The V4 attenuated scattering ratios now (correctly) show significant aerosol in the altitude region used for the V3 calibration, with a strong maximum appearing over the tropics (Figure 1). The V4 data also capture the seasonal variation of these scattering ratios (Kar et al., 2018a, see their Figure 12). This improved calibration in V4, now accurate to about 1.6%, provides the motivation for the development of the CALIPSO stratospheric product, as it enables the retrieval of aerosol extinction coefficients in regions previously (but incorrectly) assumed to be aerosol-free (Kar et al., 2018a).

## 2.2 Design and algorithm description

The level 3 stratospheric aerosol profile product reports height-resolved monthly mean profiles of aerosol backscatter and extinction coefficients on a uniform spatial grid that extends 5° in latitude (from 85°N to 85°S), 20° in longitude (from 180°W to 180°E), and 900 m in altitude. Given the low SNR in the stratospheric backscatter measurements, it is necessary to average the

data substantially, both spatially and temporally. Averaging the backscatter data over 5° in latitude increases the SNR by a factor of 40 (compared to single shot profiles) and provides a reasonable depiction of stratospheric aerosol distribution.  This is also consistent with the early results of Thomason et al. (2007), who used the early CALIPSO measurements together with data from the CALIPSO simulator (Powell, 2005) to show that averaging the data over 5° in latitude and about 1 km in the vertical resulted in fairly representative stratospheric distribution. Further, spatial distributions of stratospheric species tend to be zonally symmetric (e.g. Kremser et al., 2016). In order to capture the signature of any possible longitudinal variation, e.g., the Asian Tropopause Aerosol Layer (ATAL) which occurs over Asia every summer during the monsoon months, we have used a longitudinal grid of 20°. The altitude resolution of the CALIOP level 1 profiles varies with altitude from 60 m between 8.3 km and 20.2 km to 180 m between 20.2 km and 30.1 km and finally to 300 m between 30.1 km and 40.0 km. In order to achieve a uniform altitude resolution, the vertical grid resolution was set to 900 m. Note that the tropopause can occur below 8.3 km at high latitudes, but the vertical resolution of level 1 profiles changes again below this altitude and the lower limit was kept at 8.3 km as a trade-off between computational complexity and the stratospheric information content, while the upper limit was set at 36 km, which is the lower limit of the calibration region. The tropopause heights were taken from the Modern-Era Retrospective analysis for Research and Applications 2 (MERRA-2) reanalyses as in all V4 products (Gelaro et al., 2017). In the current version of the stratospheric aerosol product we use only nighttime data as they have significantly better SNR as compared to the daytime data (Hunt et al., 2009).

Each level 3 stratospheric aerosol file reports two distinct realizations of the monthly averaged data products. The first of these is the "background" mode, which is designed to represent the long-term background stratospheric aerosol loading. In order to achieve this, we need to remove all readily detectable perturbations within the stratosphere, such as overshooting cirrus clouds, polar stratospheric clouds (PSCs), and strongly scattering injections of smoke, volcanic ash, and other aerosol species which are detected using the layer detection algorithm implemented in the CALIOP  level 2 data processing (Vaughan et al., 2009).  The second realization is the "all aerosols" mode which is designed to represent the time history of aerosol loading in the stratosphere resulting from all possible sources.  In this case, the clouds and PSCs are still removed, exactly as is done for the background mode, however, subject to various quality assurance tests,

the aerosol layers detected in the level 2 analyses are retained. Details of the averaging algorithms and the various data filtering schemes are provided in the following sections.

### 2.2.1 Gridding and filtering

The overall design of the level 3 stratospheric aerosol product is shown in Figure 2. To begin with, three input files are required for each granule under consideration. A CALIOP granule comprises half an orbit of data either from the daytime or the nighttime part of the orbit and divided by the day-night terminator. As noted in section 1, the primary input files used for the present product are the lidar level 1B file, with the corresponding level 2 5 km merged layer and PSC mask files (Pitts et al., 2009) used for filtering. While the level 1B and level 2 merged layer files are based on V4, the currently available level 2 PSC files are based on V3. The latter is only available as a daily file and not for each granule separately. The 5 km merged layer file is a new product in V4 that reports the locations of all aerosol and cloud layers detected at both 5 km (also 20 km and 80 km) and single shot (333m) resolution (Vaughan et al., 2016).

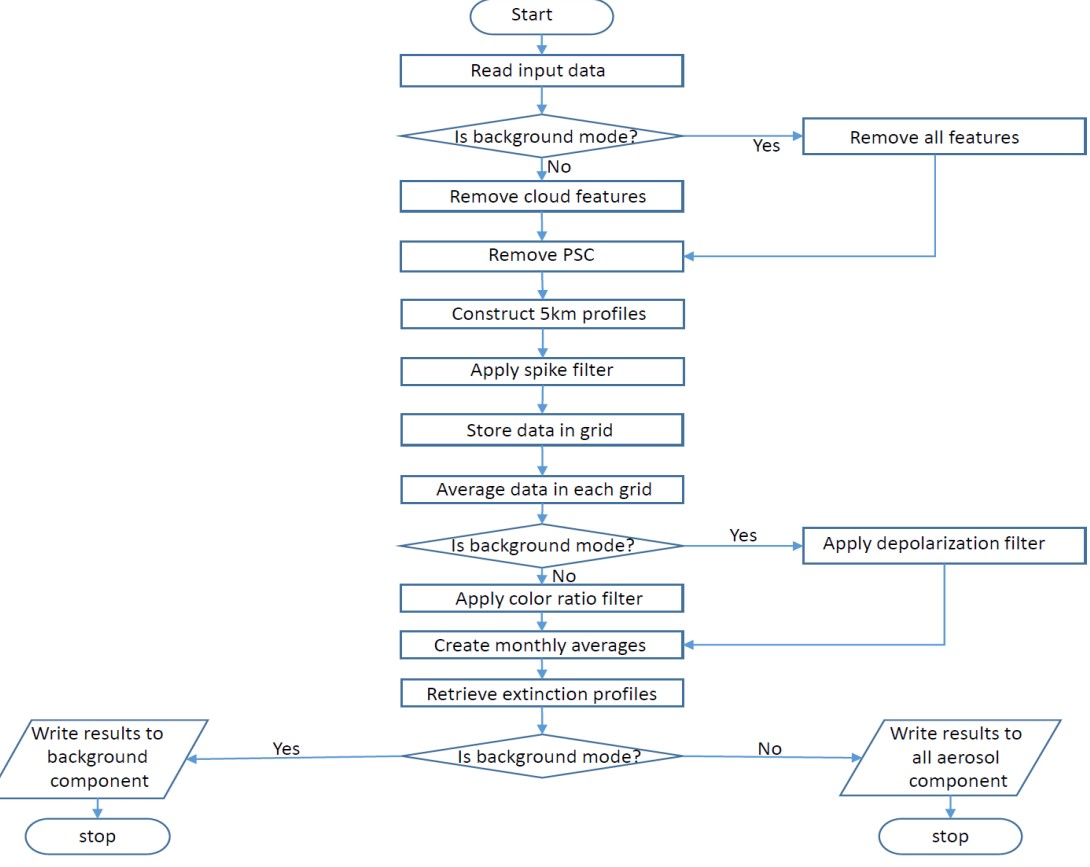

**Figure 2.** Flowchart illustrating the overall design of the CALIPSO level 3 stratospheric aerosol profile product.

In the "background mode", clearing the features detected in the level 2 analyses is done by removing all the level 1B (L1B) attenuated backscatter values (for 15 consecutive L1B profiles) beginning at the top of the uppermost cloud or aerosol layers detected above the local tropopause using the layer heights reported in the 5 km merged layer file.. Not only are signals from within the boundary of the layers removed, the backscatter values at all altitudes below the layers are also
removed to avoid issues in correcting for signal attenuation from overlying layers. While the attenuated backscattered signals within and below these layers are removed, this step will retain values which fall below the minimum detectable attenuated backscatter threshold of the CALIPSO layer detection algorithm (McGill et al., 2007). In this sense, the retrieved extinction in this mode will reflect only the aerosol loading below this threshold. Similarly, the signals below the
uppermost PSC layers are also removed using the PSC mask file for the PSC-active months in the two hemispheres (December through March in the Arctic and May through October in the Antarctic) . The PSC mask files report the occurrence of PSCs in both hemispheres (Pitts et al., 2007, 2009) and are reported for a single day on a 5 km horizontal and 180 m vertical grid for nighttime conditions only.

After clearing all level 2 and PSC layers detected above the local tropopause, all L1B attenuated backscatter values below the tropopause are removed. Further, all L1B profiles within the South Atlantic Anomaly (SAA) region are also removed. In this region, between approximately the equator and 50ºS in latitude and 20ºE to 80ºW in longitude (in the operational algorithm a polygon is used), the Van Allen belts come down to their lowest altitude ($< 200$ km), thus exposing
the satellite sensors to high fluxes of energetic charged particles which are trapped within the belts (Hunt et al., 2009, Noel et al., 2014). Large amplitude noise excursions are often observed in attenuated backscatter profiles within this area, thus degrading the already low SNR in the stratosphere. Consequently, data over the SAA are not included when calculating the level 3 stratospheric aerosol product.

When creating the all aerosol mode of the stratospheric aerosol product, it is necessary to remove any clouds and PSCs, much the same way as for the background case, but retain the

detected layers classified as aerosols by the CALIPSO cloud-aerosol discrimination (CAD) algorithm (Liu et al. 2009, 2019). It should be mentioned that the CAD algorithm was also modified in V4 in order to be compatible with the new V4 532 nm calibration (Liu et al., 2019). In fact, the CAD algorithm was extended to the stratosphere for the first time in V4. Up until V3, any layer in the stratosphere was simply classified as a "stratospheric feature" and no distinction was made between clouds and aerosols, which is no longer the case in V4. However, even the V4 CAD algorithm may not perform very well at high altitudes because of low SNR leading to generally lower absolute values of CAD scores (Liu et al., 2019). In any case, in the stratospheric altitudes above ~20 km, clouds are seldom observed (except in the polar regions) and uncertainties in the CAD algorithm are not likely to affect the stratospheric aerosol product. For this mode, only aerosol layers with acceptable CAD scores (-100 to -20) are retained within the stratosphere. Layers identified as aerosols but with unacceptably low CAD scores (between 0 and -20) are removed as if they were clouds or PSCs.

In the next step, a nominal 5 km resolution profile is constructed by taking the average of these 15 filtered L1B attenuated backscatter profiles. Subsequently, a noise filter is used to screen out strong outliers from these 5 km profiles that might otherwise lead to biases in high latitude and/or high altitude regions. The noise filter used for the current version of the product is a reconfigured version of the same filter that is used in the CALIPSO range dependent automated level 2 layer detection algorithm. Essentially a range dependent threshold array of attenuated scattering ratios is constructed, which incorporates noise from two types of sources. The first category is the range invariant noise and includes detector dark noise and noise from the solar background light. The second category is the range dependent noise from single shot measurements and is calculated from the molecular models. Using this range dependent threshold, outliers are removed (for details see Vaughan et al., 2009, section 2c). After removing the outliers, the 5 km profile is assigned to the appropriate spatial grid. This process is then repeated for all the profiles in the level 1B file. The resulting filtered 5 km profiles are then averaged to create a single mean attenuated backscatter profile for each grid cell.

In the final processing step for each granule, another quality screening is employed to identify and remove any lingering tenuous cirrus cloud in the lower stratosphere that might have escaped the layer detection mechanism due to low backscatter values. For the "background" mode, we can safely assume the background aerosols are uniformly spherical and thus have a near zero

depolarization ratio. Since ice crystals in even the most tenuous cirrus violate this assumption, we use a threshold of 5% in volume depolarization ratio (ratio of the attenuated backscatter measured in the perpendicular and parallel channels at 532 nm (Hunt et al., 2009)) to detect weakly scattering residual clouds. However, for the "all aerosol" mode, this strategy will not work. This is because volcanic ash is typically non-spherical and has high volume depolarization values (~25-30%) and thus would be removed along with the cirrus clouds. On the other hand, attenuated color ratio values (i.e., the ratio of the total attenuated backscatter coefficients at 1064 nm and 532 nm) are generally larger for clouds as compared to volcanic ash and thus may be used to filter out clouds while still retaining volcanic ash (Winker et al., 2012; Vernier et al., 2014). We have used a threshold value of 0.5 in attenuated color ratio in the "all aerosol" mode in an attempt to retain the volcanic ash rather than the filter on volume depolarization ratio. The effects of these filters in both modes are illustrated in Figure 3 using height-latitude cross sections of attenuated scattering ratios for the month of June 2011.

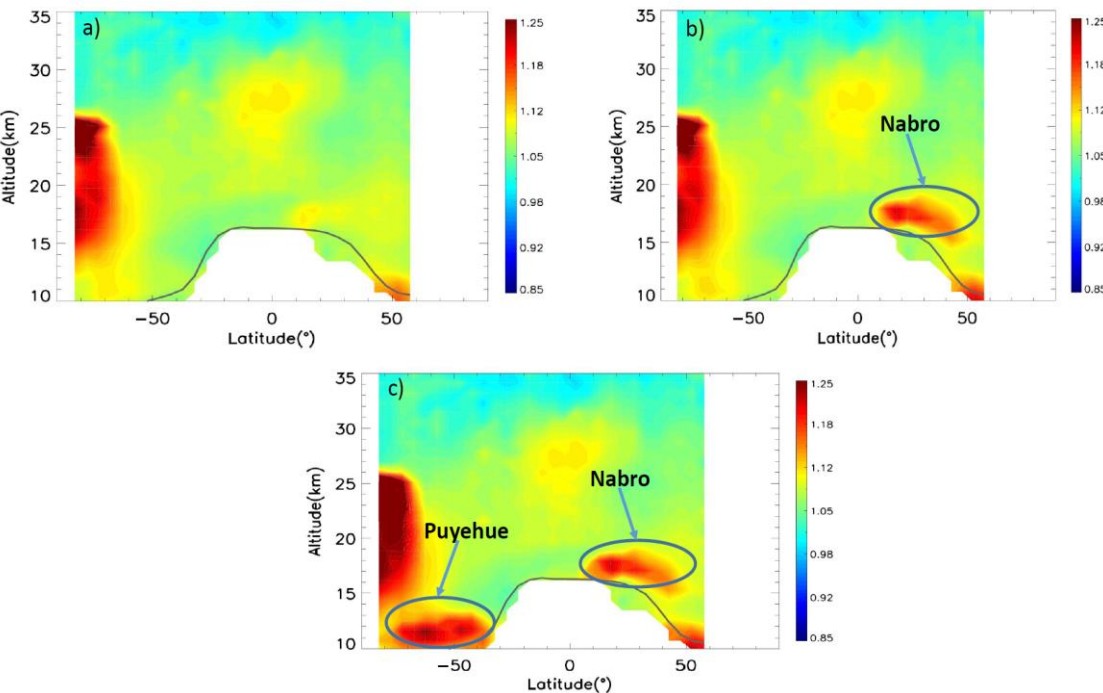

**Figure 3.** Zonally averaged height-latitude cross sections of attenuated scattering ratio for June 2011: a) after removing all detected layers and using a volume depolarization ratio filter (i.e., background aerosol only); b) including aerosol layers

in the stratosphere detected by the level 2 algorithms with a 5% volume depolarization ratio filter applied; and c) including the level 2 aerosol layers but using an attenuated color ratio filter instead of the volume depolarization ratio filter. The white area in the northern high latitudes in summer indicates lack of nighttime data.

During this month two strong volcanic eruptions took place, Nabro in the northern hemisphere (June 13th, 13°N, 41°E) and Puyehue-Cordon Caulle in the southern hemisphere (June 4th, 40°S, 72°W). The composition of the Nabro plume was mostly sulfate while the composition of Puyehue-Cordon Caulle was mostly ash, at least initially (de Vries et al., 2014; Vernier et al.,
2013). In the background mode (Figure 3a), removal of all detected layers combined with the application of the volume depolarization filter ensures that stratospheric perturbations from these two volcanoes are mostly excluded. Figure 3b shows the effect of including aerosol layers in the stratosphere with acceptable CAD scores ($|CAD| > 20$) while still using the volume depolarization filter. Now the Nabro plume can be clearly seen but not that of Puyehue-Cordon Caulle. This is
because the sulfates in the Nabro plume have low volume depolarization ratios that fall below the threshold and are thus retained while the ash layers with high volume depolarization ratio from Puyehue-Cordon Caulle are removed. On the other hand, including the aerosol layers but substituting an attenuated color ratio threshold of 0.5 in place of the volume depolarization ratio filter, as shown in Figure 3c (all aerosol mode), reveals both the Nabro (near 30°N) and Puyehue-
Cordon Caulle plumes (near 50°S) quite clearly. Note the high scattering ratio values in the Antarctic latitudes between 15-km and 25-km. Because the PSC mask algorithm is optimized specifically for PSC detection, its increased sensitivity allows it to detect a considerably larger fraction of faint PSCs relative to the more generalized and generic level 2 feature detection algorithm. Since all PSC layers detected by the dedicated PSC detection algorithm were removed,
what remains are the signatures of only those particles below the detectability threshold of the PSC mask data product. Note that the enhanced scattering ratios near 25-30 km represent the tropical reservoir of stratospheric aerosols (Trepte and Hitchman, 1992, Kremser et al., 2016). Further, the high scattering ratios near 50°N are likely due to the Grímsvötn volcano, which erupted in May 2011.
Using a constant threshold to discriminate between different classes of inherently noisy measurements can entail significant risk of misclassification. For example, using a higher

attenuated color ratio acceptance threshold to ensure the identification of strong ash plumes (e.g., for Pueyhue Cordon Caulle above) may result in a significant amount of cloud contamination. Similarly, an acceptance threshold set too low will likely exclude all clouds while simultaneously discarding much of the ash signal.

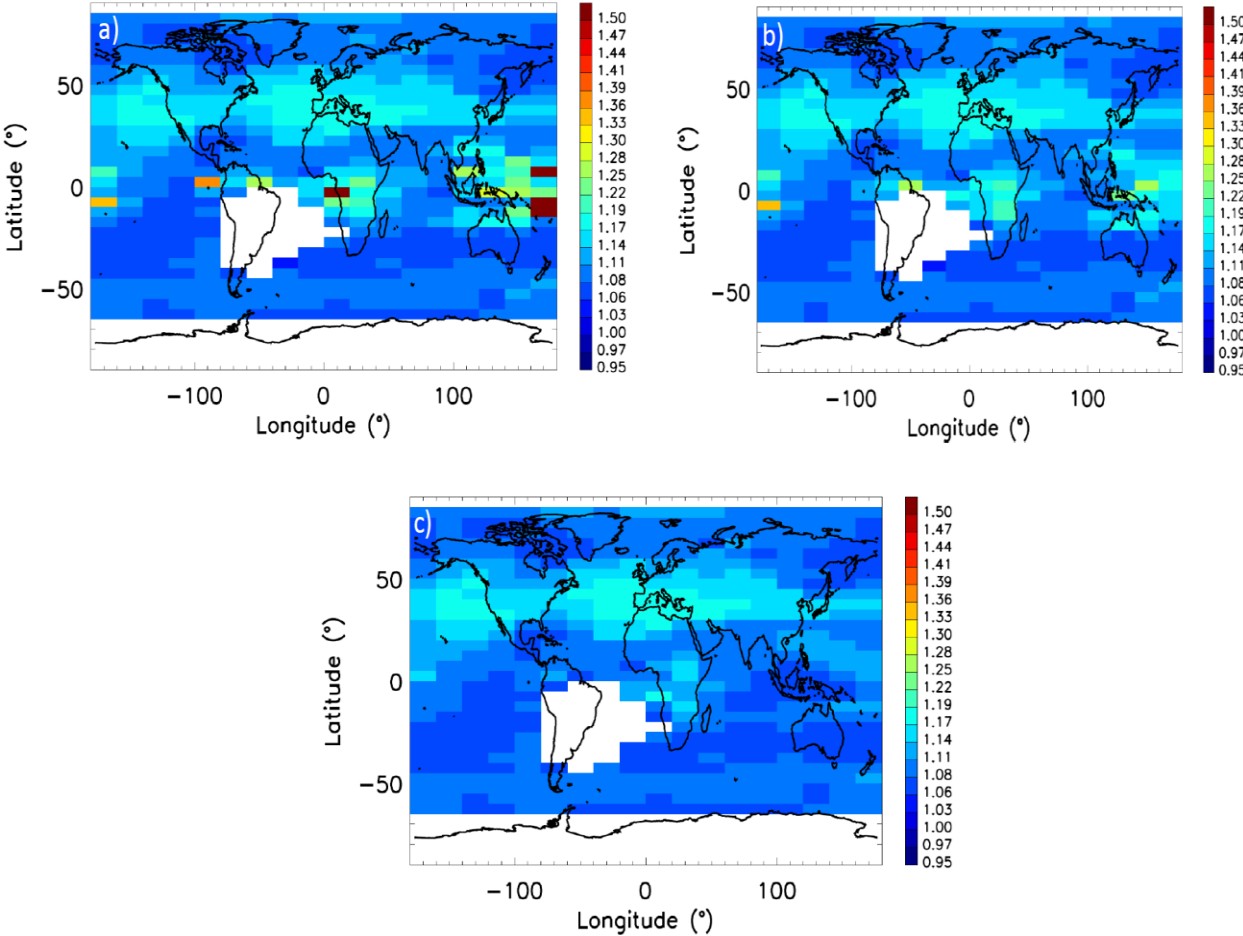

**Figure 4**: Attenuated scattering ratios at 17 km in December 2011, a) including all detected
aerosol layers but before applying any filter to remove the thin cirrus clouds, b) including all detected aerosol layers after applying the filter in attenuated color ratio (the "all aerosol" mode) and c) excluding the detected layers and after applying the filter in volume depolarization ratio (the "background" mode).

The impact of using the attenuated color ratio and volume depolarization ratio filters on removing thin cirrus clouds in, respectively, the all aerosol and background only components is illustrated in Fig. 4 using the attenuated scattering ratios measured at 17 km during December 2011. In Fig. 4a, all the aerosol layers are retained, much like the "all aerosol" component, except that neither the volume depolarization ratio filter nor the attenuated color ratio filter is used. The high scattering ratios between about 30°N-55°N are due to the Nabro plumes which have spread around the northern hemisphere by December 2011. Apart from this band, high scattering ratios are also seen in a tropical band (between 25°S to 25°N) over the Western Pacific as well as over parts of Africa. These reflect the thin cirrus clouds occurring in the upper troposphere and lower stratosphere near the tropopause (Sassen et al., 2009). Figure 4b shows the distribution in the "all aerosol" mode where the attenuated color ratio filter is used. Clearly a significant number of pixels with high scattering ratio (thin cirrus clouds) in the tropics has been removed while still retaining the volcanic aerosol signature. In Figure 4c we see the impact of the volume depolarization filter in the "background" mode. Now most of the cirrus clouds have been removed. Note that the aerosol signature has remained much the same in all three distributions. That is because by December 2011, there may not be many volcanic layers left, as such, yet enhanced scattering from the volcanic material is still present. This figure shows that for quiescent conditions or when the aerosol load is not very high (so that not many plumes are detectable as layers in CALIPSO L2 algorithm), the volume depolarization filter will do a better job of clearing the thin cirrus clouds, thus making the background component as the mode of choice. In any case, note that the thin cirrus clouds mostly affect the tropical latitudes and near the tropopause ~16-18 kms (Sassen et al., 2009). Note that both the volume depolarization ratio and attenuated color ratio filters are applied only below 25 km, as no cirrus is expected above this altitude.

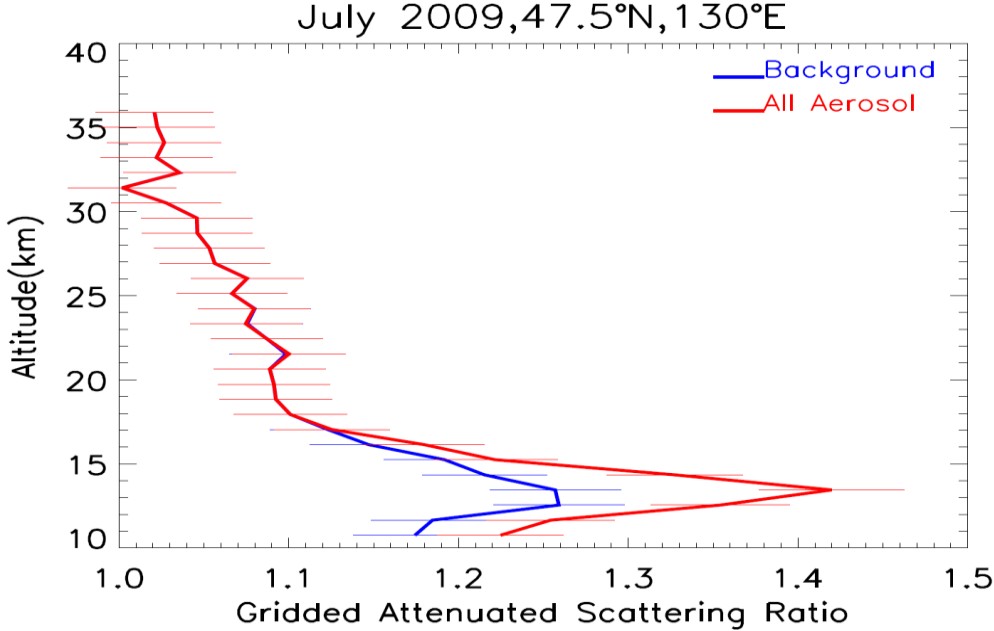

**Figure 5.** Profiles of attenuated scattering ratio at 47.5ºN and 130ºE in July 2009 for the background (blue) and all aerosol (red) components. The error bars represent computed uncertainties.

Figure 5 shows the profiles of attenuated scattering ratio for the background and all aerosol modes in July 2009 for the grid cell centered at 47.5ºN and 130ºE. The enhanced scattering ratio in the lower stratosphere between 10-km and 17-km is due to the inclusion of detected aerosol layers from the Sarychev volcano (48.1ºN, 153.2ºE), which erupted in June 2009. Note that backscatter from some of the Sarychev aerosols which fall below the minimum detectable backscatter threshold of the level 2 layer detection algorithm will contribute to the background profile.

After deriving the granule-averaged data, we create monthly averaged gridded profiles of attenuated backscatter by aggregating all profiles during each month of the mission. In addition to the attenuated backscatter coefficient profiles, profiles of molecular and ozone number densities, temperatures, and pressures reported in the L1B files are also averaged and gridded for use in the subsequent retrieval procedures.

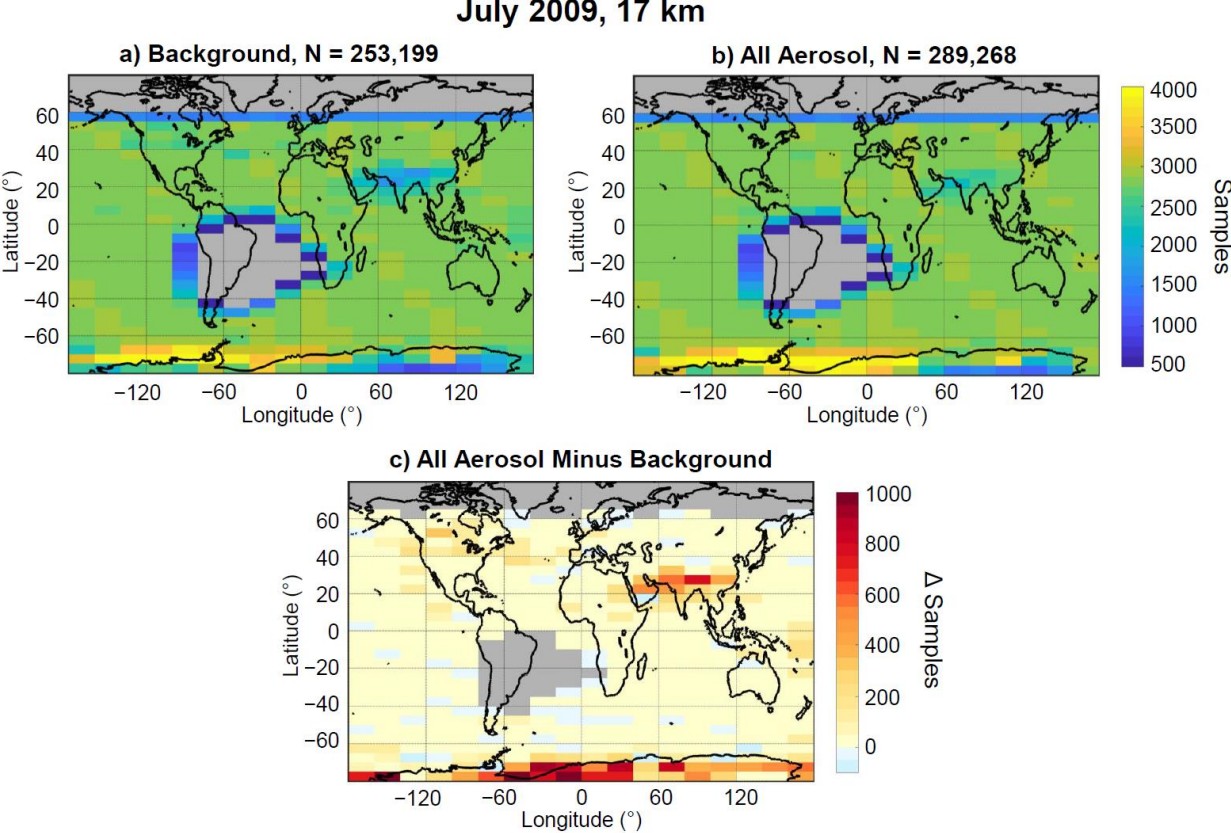

**Figure 6**. Number of samples contributing to a) background mode and b) all aerosol mode and c) the difference between the two modes at 17 km in July 2009. Grid cells with < 50 samples are plotted in grey.

Figure 6 depicts the spatial distribution of the number of samples that contributed to the two components at 17 km during July 2009. The grey grid cells over South America and parts of South Atlantic Ocean correspond to the SAA, over which all data samples are rejected. –The higher number of samples for the all aerosol mode over the Asian summer monsoon region reflects the signature of the aerosol in the Asian Tropopause Aerosol Layer (ATAL, Vernier et al., 2011b). Higher number of samples in the all aerosol mode, albeit on a lesser degree, can also be seen over North America, which is likely related to the Sarychev volcano as mentioned above. Also note the high number of samples over parts of Antarctica, partly from oversampling due to orbital configuration and related to small particles below the detectability of PSCs by the PSC mask algorithm.

**2.2.2 Retrieval of aerosol extinction profiles**

The monthly mean profiles of gridded 532 nm attenuated backscatter coefficient (β′), constructed using the procedure described in the preceding section, along with gridded profiles of molecular backscatter coefficients ($\beta_m$), molecular extinction coefficients ($\alpha_m$), and ozone absorption coefficients ($\alpha_{O_3}$), are used to retrieve the particulate backscatter coefficient ($\beta_p$) using

$$\beta_p(z) = \beta'(z) \, / \, T_m^2(z) \; T_{O_3}^2(z) \; T_p^2(z) - \beta_m(z) \tag{2}$$

where,

$$T_m^2(z) = \exp\!\left(-2 \int_0^z \alpha_m(r') \, dr'\right), \tag{3a}$$

$$T_{O_3}^2(z) = \exp\!\left(-2 \int_0^z \alpha_{O_3}(r') \, dr'\right), \text{ and} \tag{3b}$$

$$T_p^2(z) = \exp\left(-2 \, \eta_p \, S_p \int_0^z \beta_p(r') dr'\right). \tag{3c}$$

In these expressions, $\eta_p$ is the particulate multiple scattering factor, $S_p$ is the particulate lidar ratio (i.e., the extinction-to-backscatter coefficient ratio), and $T_m^2(z)$, $T_{O_3}^2(z)$, and $T_p^2(z)$ are, as previously defined, the molecular, ozone, and particulate two-way transmittances. The molecular

backscatter coefficients and molecular and ozone two-way transmittances can be calculated from molecular model data (e.g., as described in Kar et al., 2018a). The molecular model used exclusively throughout the CALIPSO V4 data products is MERRA-2 provided by NASA's Global Modeling and Assimilation Office (Gelaro et al., 2017). For the CALIPSO stratospheric aerosol product, the particulate multiple scattering factor is taken as 1 for all species of stratospheric

aerosols, consistent with the approach taken in the CALIPSO level 2 aerosol retrievals (Young et al., 2013; Young et al., 2016; Young et al., 2018).

Given an appropriate value of the lidar ratio, equations 2 and 3c can be solved iteratively to obtain estimates of $\beta_p(z)$ (Young and Vaughan, 2009). Estimates of particulate extinction coefficients are subsequently obtained using $\sigma_p(z) = S_p \times \beta_p(z)$. The V1.00 release of the level 3

stratospheric aerosol product uses a value of $S_p = 50$ sr for the stratospheric aerosol lidar ratio, which is a typical value used for stratospheric aerosols for background conditions and in absence of significant ash and sulfate injections from volcanoes (Trickl et al., 2013, Ridley et al., 2014, Sakai et al., 2016, Kremser et al., 2016, Khaykin et al., 2017). Note that the lidar ratios could also be significantly different for stratospheric perturbations resulting from smoke intrusion from

pyroCb events (Peterson et al., 2018, Khaykin et al., 2018). For this first version of CALIPSO

stratospheric aerosol product we have used only a single lidar ratio. We also assume $S_p$ to be constant at all latitudes and over the entire altitude range. The retrievals are carried out beginning at 36 km and extending downward to either 8.3 km or 1 km below the tropopause; processing stops when the higher of these two altitudes is reached. Extending the range below the tropopause, when

possible, is intended to help in studies of the upper troposphere and lower stratosphere (UTLS).To guarantee uniform results across multiple CALIPSO data product levels, the level 2 CALIPSO extinction retrieval module is used to calculate the level 3 profiles of stratospheric aerosol extinction and backscatter coefficients and their uncertainties. The details of the retrieval process and uncertainty estimates are given in Young and Vaughan (2009) and Young et al. (2013, 2016,

2018) and are not repeated here.

### 3. Initial assessment of CALIPSO stratospheric aerosol product

In this section we assess the initial performance of the CALIPSO stratospheric aerosol product by first presenting the signatures of various stratospheric aerosol events as captured by the product and then making quantitative comparisons with observations from SAGE III on ISS.

### 3.1. Signatures of stratospheric events and dynamics.

### 3.1.1 Effects of volcanic and smoke injections.

Volcanoes are one of the primary sources of stratospheric aerosols (e.g. Kremser et al., 2016). Ground based lidar studies have indicated a positive trend in the stratospheric sulfate aerosol

loading since the turn of the century which was initially attributed to anthropogenic emissions of $SO_2$ from coal burning in South East Asia (Hofmann et al., 2009). However, closer scrutiny suggests that the increase is instead related to emissions of $SO_2$ from a large number of moderate volcanic eruptions, as was subsequently suggested by Vernier et al. (2011a) based on analyses of CALIPSO data. Several volcanoes with stratospheric impacts have been recorded since the study

by Vernier et al. (2011a). Volcanic signatures in CALIPSO data were examined more recently by Friberg et al. (2018).

Figure 7 shows the time-altitude cross section of the zonally averaged extinction in the mid-high northern latitude (40°N-60°N), tropics (25°N-25°S), and mid-high southern latitude (40°S-60°S) from the CALIPSO level 3 stratospheric aerosol product between January 2007 and

December 2017. The signatures of many volcanoes are clearly evident in this figure in all three latitude bands. The strongest extinctions are seen for Sarychev and Nabro in the northern hemisphere and Calbuco in the southern hemisphere. Note that the effects of some of the volcanoes can last for several months as they spread to other latitudes by isentropic transport.

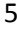

**Figure 7.** Time-altitude cross sections of the retrieved extinction coefficients in all aerosol mode from January 2007 through December 2017 for a) mid-high northern latitudes, b) the tropics and c) mid-high southern latitudes. The white areas indicate missing data. Note that in the middle panel for the tropics, the altitude ranges from 17 km to 25 km, whereas the range is 12 km to 25 km in panels a) and c).

Apart from volcanic material, smoke from strong biomass burning events can also reach the stratosphere during so-called pyrocumulonimbus events (Fromm et al., 2010; Peterson et al., 2018). During the "Black Saturday" event, smoke from strong bushfires in Victoria, Australia on February 7, 2009 is known to have impacted the stratosphere. Plumes from this blaze eventually reached altitudes of 16–20 km and were readily visible in satellite imagery (de Laat et al., 2012; Glatthor et al., 2013). The signature of this event can also be identified in Figure 7b, reaching up to nearly 22 km. The signature of another strong pyroCb event can be seen at northern mid-high latitudes (top panel) in August-September, 2017. This event is discussed in detail below. Note the seasonal pattern of high extinction near 12-15 km in northern mid-high latitude summer, seen most clearly between 2012 and 2017. The reason for this is not entirely clear at this time, but could again be due to fire events. Cirrus cloud contamination could be another factor. However, the same pattern is also seen in the "background" mode (not shown) to a slightly lesser degree which suggests cloud contamination may not be very significant.

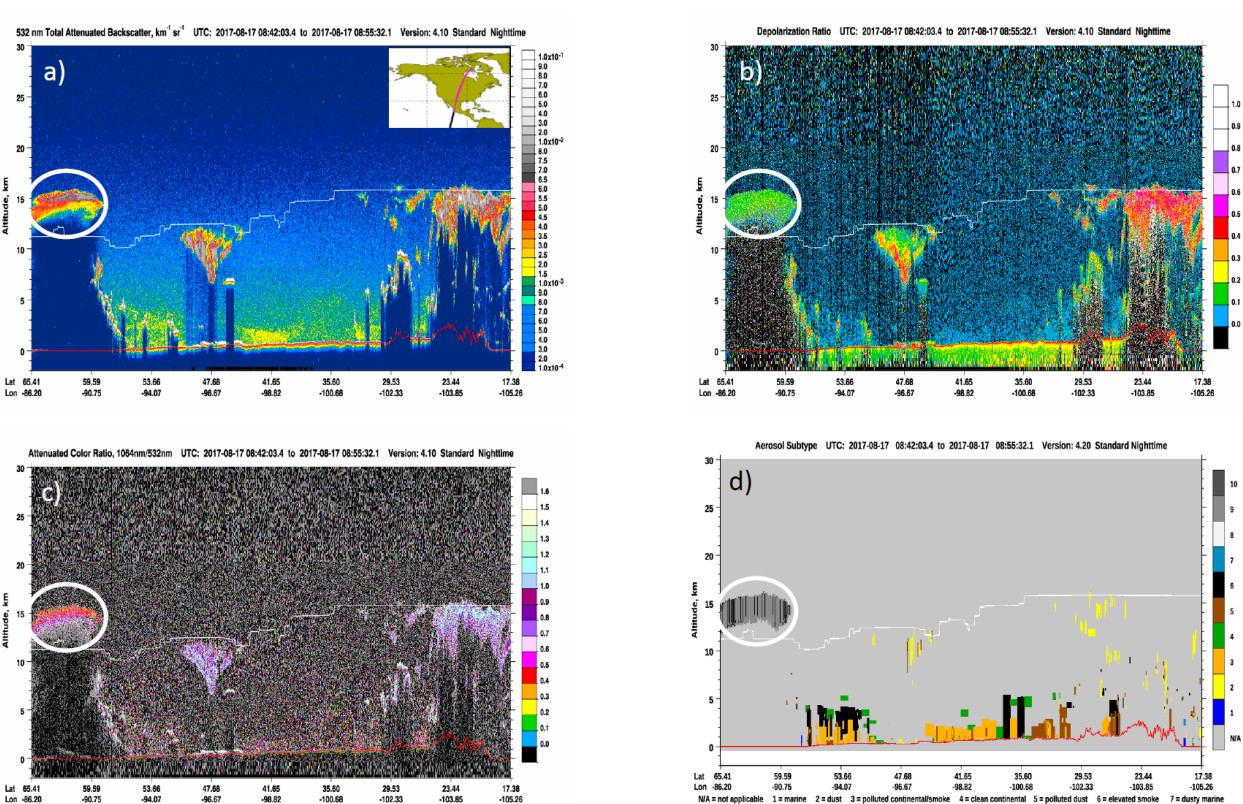

**Figure 8**. CALIPSO browse images of a) 532 nm total attenuated backscatter, b) 532 nm volume depolarization ratio, c) attenuated backscatter color ratio (1064 nm / 532 nm) and d) aerosol subtypes for a pyroCb event over Canada on August 17, 2017. The smoke plume is shown in the white circles. The white lines indicate the MERRA-2 tropopause altitude.

The extreme pyroCb event that occurred in August 2017 over British Columbia in Canada has been extensively studied recently and has been likened to volcanic perturbations in the stratosphere in terms of intensity and duration (Khaykin et al., 2018; Ansmann et al., 2018; Haarig et al., 2018, Peterson et al., 2018). Figure 8 shows an example of the CALIPSO measurements of this pyroCb

event. The signature of the smoke plume is seen as extremely high attenuated backscatter (opaque at 532 nm) between 60°N and 65°N. The very high attenuated color ratio (~1.6) seen at the base of the plume (Figure 8c) is a tell-tale signature of smoke (e.g. Liu et al., 2008). The high volume depolarization ratio ($\geq$ 0.1) seen in Figure 8b is somewhat unusual for smoke and suggests the presence of irregular soot particles and mineral dust and possibly some ice particles, with fast

adiabatic lifting  possibly retaining the initial irregular shapes (Haarig et al., 2018, Khaykin et al., 2018). This high color ratio combined with the unusually high depolarization results in the plume being identified as a mixture of smoke and "volcanic ash" (Figure 8d), the latter being misclassifications by the CALIOP V4 level 2 scene classifier.

Figure 9 shows the height-latitude cross sections of CALIOP attenuated scattering ratios

from the stratospheric aerosol product between August 2017 and November 2017 and captures the evolution of the aforementioned pyroCb event. After the original injection of smoke in August 2017 at mid-latitudes, the smoke spreads to lower latitudes as can be seen in these monthly mean spatial distributions from the level 3 stratospheric aerosol product.  As in Figure 3, the feature with high attenuated scattering ratio near 25-30 km seen in all the four panels is the signature of the

tropical reservoir of stratospheric aerosols, maintained by a complex interplay of transport from the troposphere and stratospheric dynamics as well as microphysical processes including the Brewer-Dobson circulation, the QBO, evaporation and sedimentation (Trepte and Hitchman, 1992, Kremser et al., 2016).

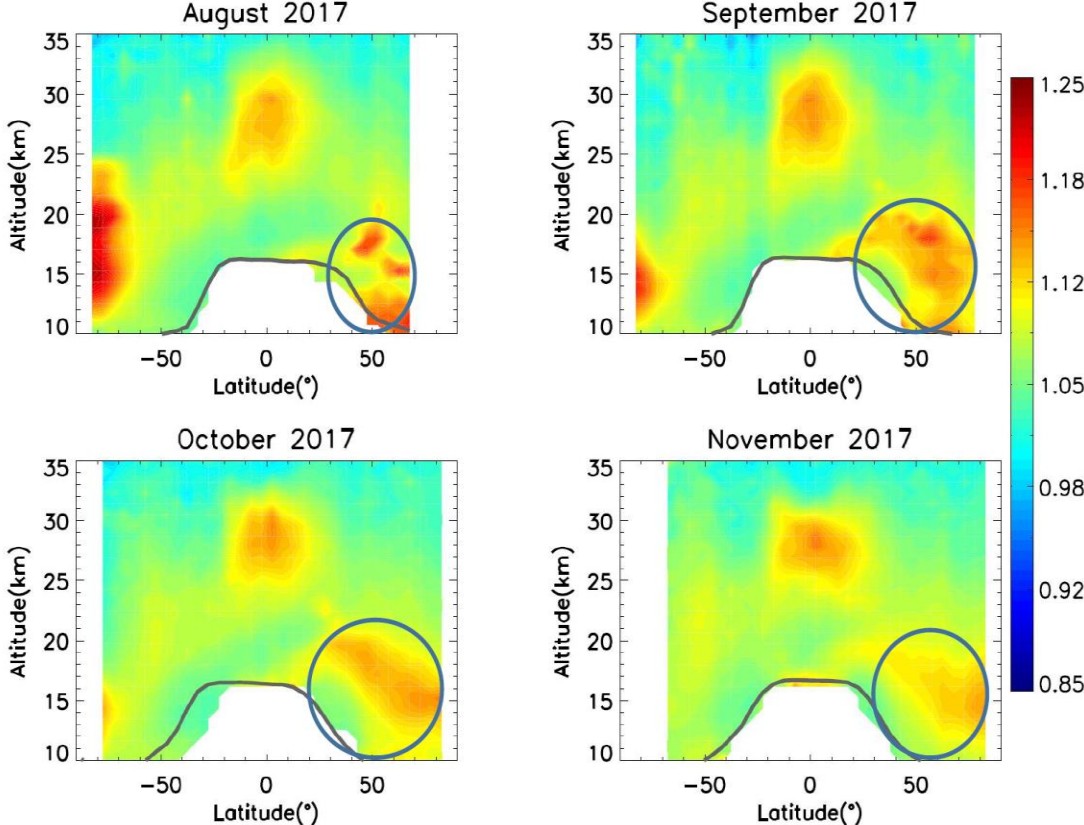

**Figure 9**. Zonally averaged height-latitude cross section of the 532 nm attenuated scattering ratios from August 2017 through November 2017. The white areas in the northern high latitudes in August and the southern high latitudes in November indicate the lack of nighttime data due to continually changing day-night terminator times.

### 3.1.2. Signatures of stratospheric dynamics

Figure 10 shows the height-latitude cross section of the retrieved 532 nm extinction coefficients for the all aerosol mode from March to December 2014, which captures the evolution of the Kelud eruption (February 2014, 7.9ºS, 112.3ºE) in altitude. The gradual lofting of the plume, with its top rising from ~21 km over the tropics in March to ~24 km in the same general location several months later, shows the signature of stratospheric dynamics in the CALIPSO stratospheric aerosol product. The persistence of the stratospheric perturbation for several months is consistent with the results of Vernier et al. (2016) who found the presence of ash in the lower stratosphere 3 months after the Kelud eruption from balloon observations. Note the high extinction values near

50°N-60°N in the lower stratosphere (~10-15km). These are similar to the summer rise in extinctions at these latitudes as discussed earlier (Figure 7) and are possibly due to biomass burning effects but could also be related to possible cloud clearance issues. As also mentioned above, the high extinctions at high southern latitudes could be related to scattering from particles below the PSC detection threshold as well as to transported volcanic material from Kelud.

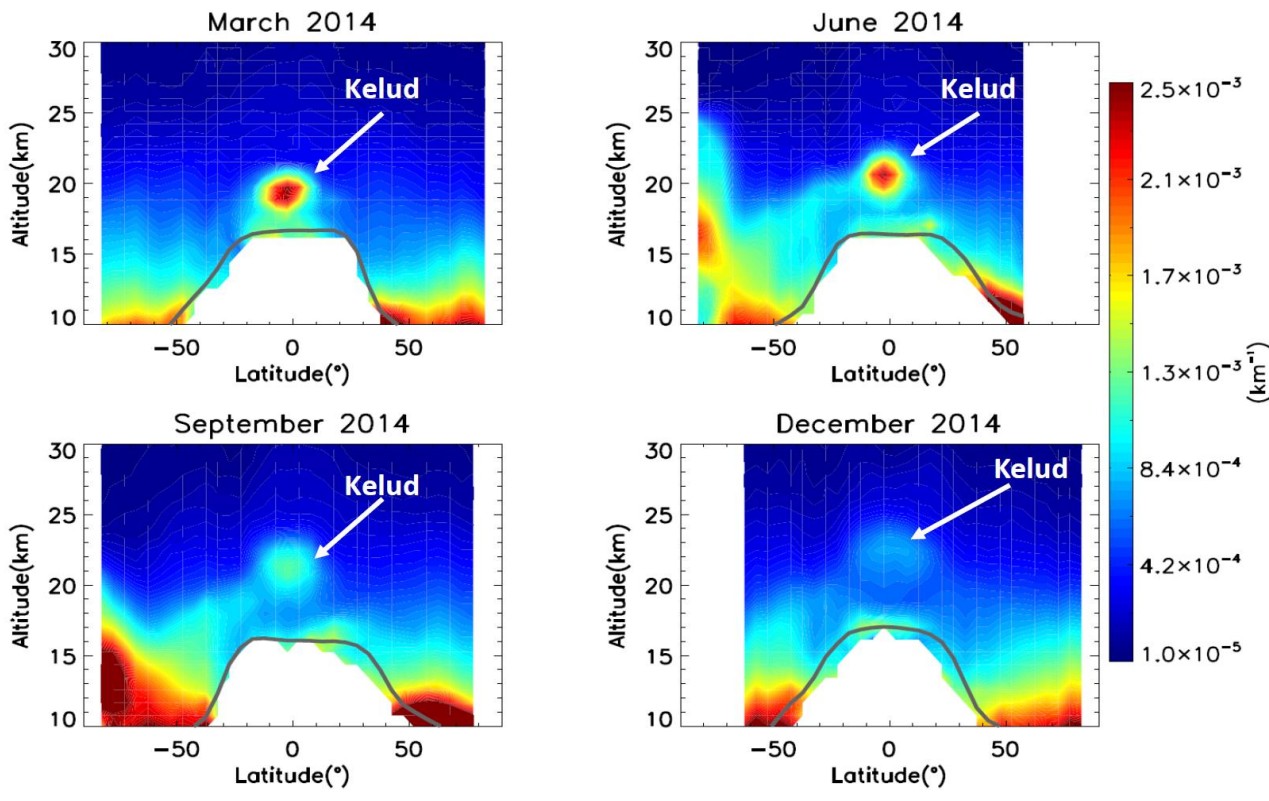

**Figure 10.** Zonally averaged height-latitude cross sections of 532 nm extinction coefficients (km$^{-1}$) in March, June, September and December of 2014. The white area in the northern high latitudes in June and in the southern high latitudes in December indicates lack of nighttime data.

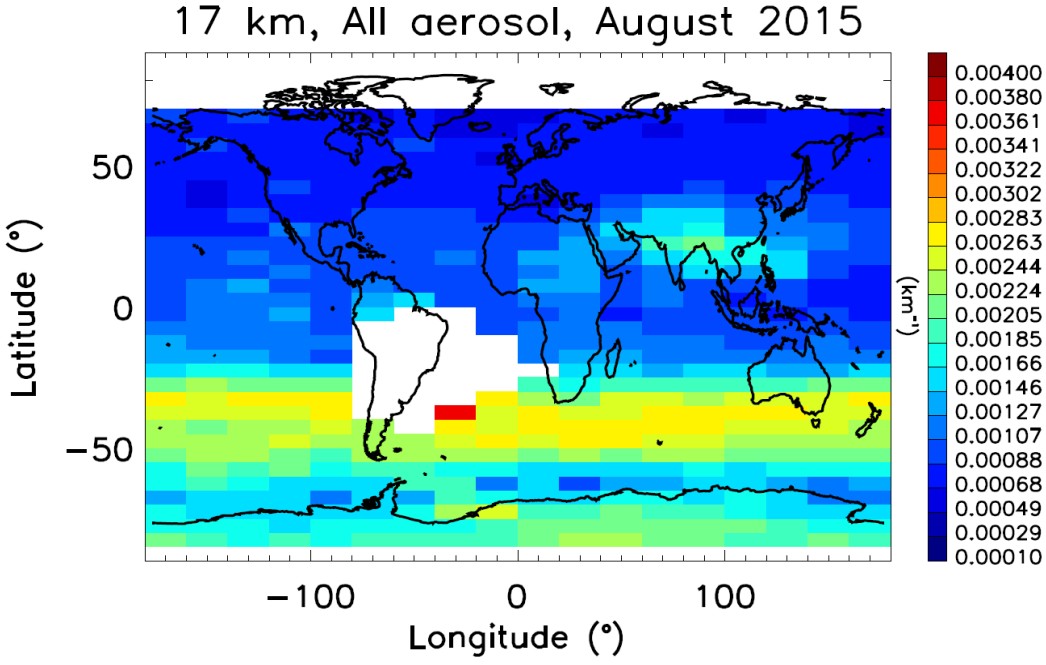

**Figure 11**. Retrieved 532 nm extinction coefficients (km$^{-1}$) at 17 km for August 2015.

Figure 11 shows the spatial distribution of the retrieved extinction coefficients at 17 km for the month of August 2015 for the all aerosol mode. Two strong perturbations of the lower stratosphere can be seen in this plot. The first is the plume from the Calbuco volcano in Chile which erupted in April 2015. The initial plumes would be missed out in the level 3 stratospheric aerosol product because data over the SAA region were not included. However the plumes quickly spread around the southern hemisphere in a belt between 60ºS to 30ºS (Lopes et al., 2019) and can be seen in the level 3 stratospheric aerosol product from May 2015 onwards for several months. The other is the plume of high extinction over southeast Asia and extending to the Arabian peninsula to the west. This is the location of the Asian summer monsoon anticyclone which has been known to be a reservoir of pollution during the monsoon months and results from deep convective outflow of pollutants both gases and aerosols and their precursors from the surface layers (Kar et al., 2004, Vernier et al., 2011b).

**3.2. Comparison with SAGE III on ISS**

In this section we provide an initial quantitative assessment of the CALIPSO level 3 stratospheric aerosol product by inter-comparison of the retrieved extinction coefficients with those from the SAGE III instrument aboard ISS (SAGE III-ISS). The SAGE III-ISS instrument was launched in February 2017 and has been providing measurements of ozone, $NO_2$, water vapor, and aerosols from its mount on the exterior of the ISS since March 2017 (Cisewski et al., 2014). The instrument derives its legacy from the long line of SAGE instruments which have been providing the most accurate retrievals of aerosol extinction in the stratosphere since 1984 (Chu et al., 1993; Thomason et al., 2008, 2010, Damadeo et al., 2013). SAGE III-ISS performs solar and lunar occultation measurements as the ISS orbits the Earth and covers a broad latitude band (60°S to 60°N) and longitude range (180°W to 180°E). The aerosol extinction profiles are available from the solar occultation measurements in 9 channels from 384 nm to 1544 nm starting June 2017. We use the latest version 5.1 extinction profiles reported in the 521 nm channel, which is closest to the CALIPSO 532 nm channel. In order to compare with the CALIPSO level 3 product, which reports gridded monthly averages, we average the daily data from SAGE III-ISS onto the same latitude grid (zonally averaged) as CALIPSO over a month and interpolate to the CALIPSO altitude grid. Data from both the sunrise and sunset occultations are used in the comparisons. Further, the data were filtered for cloud contamination by selecting only those data having a 521 nm to 1022 nm extinction ratio greater than 2 (Thomason and Vernier, 2013). We convert the SAGE III-ISSdata at 521 nm to 532 nm by using an Angstrom exponent (binned into 5° latitude bins and interpolated to CALIPSO altitude grid) derived from the extinctions retrieved at 521 nm and 1022 nm by SAGE III-ISS for the same month. Measurements from both the instruments from June 2017 through August 2018 were used for this comparison. The globally averaged value of the Angstrom exponent (between 521 nm and 1022 nm) derived using all 15 months of data is ~1.56, essentially the same as the constant value used by Khaykin et al. (2017) to convert SAGE II extinctions at 525 nm to 532 nm. We used only extinction values with corresponding fractional extinction uncertainty less than 100% for retrievals from both the instruments and calculate the differences between CALIPSO and SAGE III-ISS from the following equation:

$$\Delta (z) = 100 \times (\sigma (z)_{CALIPSO} - \sigma (z)_{SAGE}) / \sigma (z)_{SAGE} \qquad (4)$$

where $\sigma(z)_{CALIPSO}$ is the extinction coefficient at altitude z from CALIPSO and $\sigma(z)_{SAGE}$ is the extinction coefficient from SAGE III-ISS at the same altitude. Further, we use zonally averaged (into 5° latitude bins) profiles for height resolved comparisons and only the "all aerosol" mode from CALIPSO product for the sake of compatibility.

a)

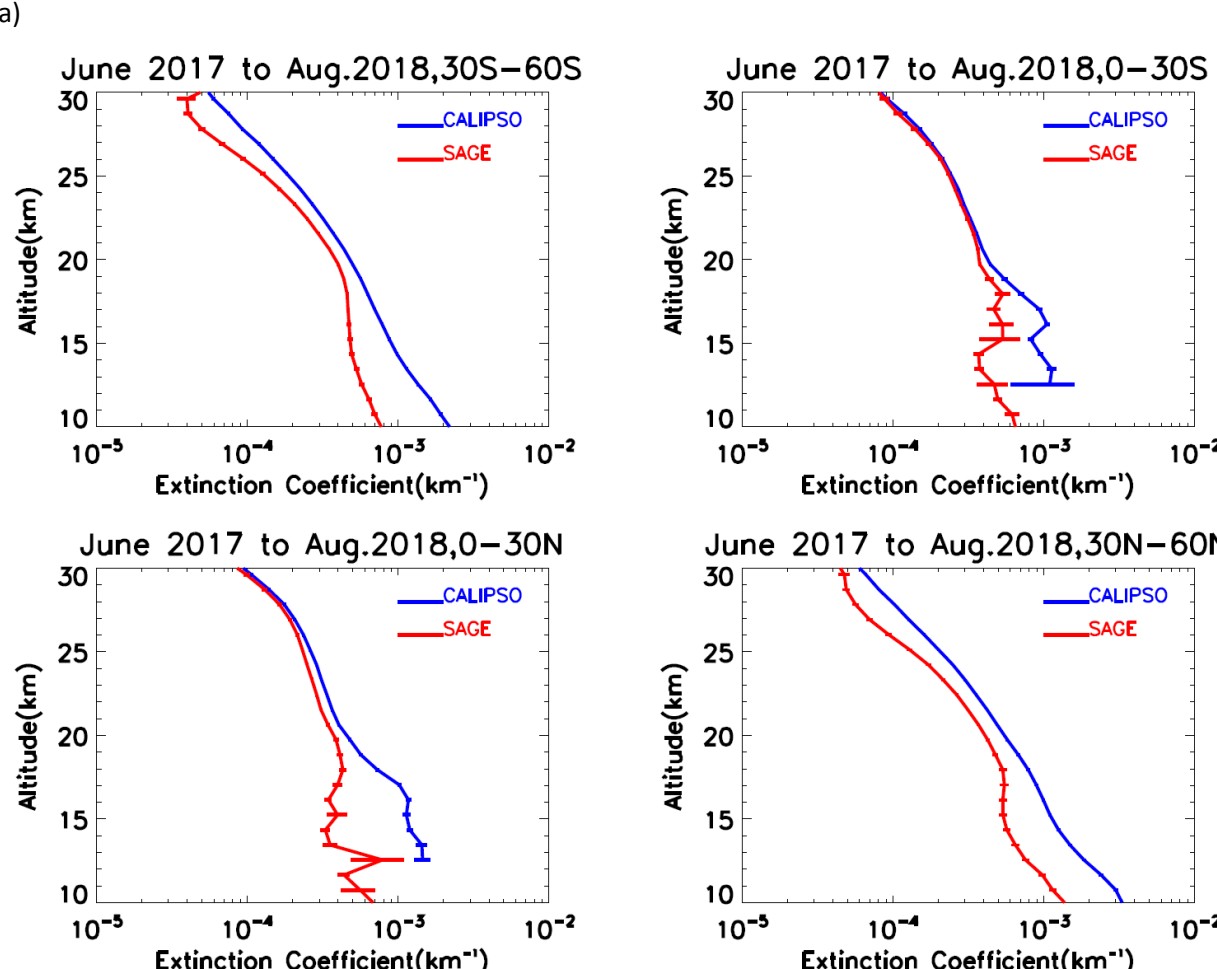

b)

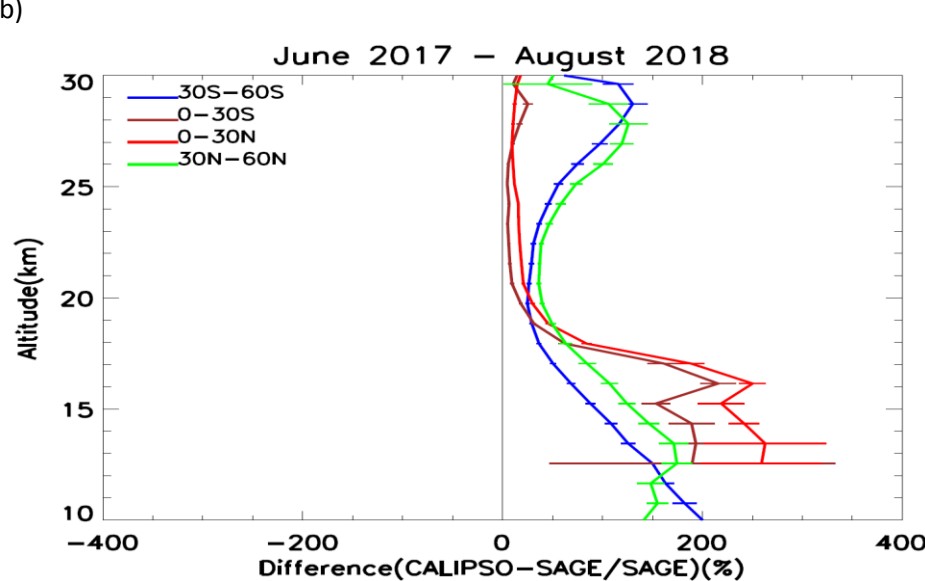

**Figure 12**    a) Altitude-resolved profiles of the mean 532 nm extinction coefficients retrieved from CALIPSO and SAGE III-ISS using all available data between June 2017 and August 2018. The corresponding average differences are shown in panel b). The differences are calculated at each latitude and altitude grid for each month and then the average is taken over all the available months. The error bars represent the standard errors of the mean.

Figure 12a shows zonally averaged mean profiles of extinction coefficients retrieved from CALIPSO (in blue) and SAGE III-ISS (in red) at 4 latitude bands using the same 15 months of measurements from the two instruments. Figure 12b shows the profiles of the fractional differences in the same latitude bands. The profiles for 0-30ºS and 0-30ºN, generally show fairly good agreement with the average difference within about 25% between 20 km and 30 km. The comparisons for 30ºS-60ºS and 30ºN-60ºN are similar and both show significant differences between CALIPSO and SAGE III-ISS extinction  with CALIPSO having a high bias of less than 50% near 20 km increasing to ~120% around 28 km. All the profiles diverge significantly at altitudes below 20 km, with the average difference often exceeding 100% and CALIPSO consistently overestimating SAGE III-ISS. It is likely cloud removal artifacts in both the instruments are affecting these lower stratospheric comparisons. As pointed out in section 2.2.1, the filtering scheme that removes thin cirrus clouds in the all aerosol mode is not as efficient as the technique employed in the background mode.  Consequently, scattering artifacts from

undetected subvisible cirrus are more likely to appear in the all aerosol mode in the tropical lower stratosphere within a few kilometers above the tropopause. Using the extinction profiles from the background mode reduces the differences at these altitudes but does not completely eliminate them (not shown). Also, note that the aerosol retrievals from SAGE III-ISS(as for the legacy retrievals

from SAGE II) are not directly filtered for the presence of clouds which may impact the retrievals in the lower stratosphere. Thomason and Vernier (2013) discuss the difficulties involved in cloud identification and clearing of the SAGE II measurements and conclude that it is not always to possible to completely eliminate cloud contamination of the aerosol extinction retrievals. Following their recommendations, we have attempted to remove the cloud contamination in the

extinction retrievals by using only those data for which the ratio of extinctions at 521 nm and 1022 nm is greater than 2. SAGE III aerosol extinctions have not been validated as of now and it is not clear if there are any issues with the retrievals at lower altitudes near the tropopause. We further discuss the possible issues resulting from uncertainties in lidar ratios below.

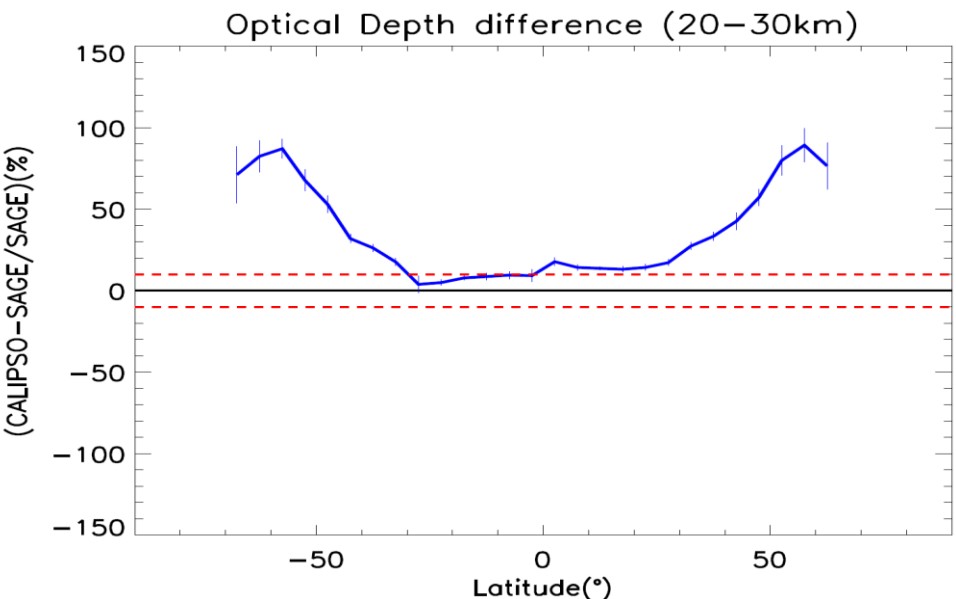

**Figure 13.** Fractional difference in 532 nm optical depth between CALIPSO and SAGE III-ISS calculated using extinction coefficients from 20-30 km, as a function of latitude.

The dashed red lines demarcate the ±10% difference levels. The error bars represent the standard error of the mean.

Figure 13 shows the difference in the stratospheric optical depths between CALIPSO and SAGE III, calculated using the average extinction coefficient profiles between 20 km and 30 km. This region is not likely to be affected significantly by clouds and also is the region where most of the stratospheric aerosol resides, thus comparisons here are likely to be indicative of the overall performance differences between the two sensors. Between 30°S to 30°N the optical depths are in agreement to within about 10-20%, though the differences begin to rise substantially in the mid-latitudes of both hemispheres.

## 4. Discussion

For an initial assessment of the CALIPSO stratospheric aerosol product, we have used the aerosol retrievals from SAGE III-ISS acquired between June 2017 and August 2018. The solar occultation technique used for SAGE III-ISS retrievals does not rely on any assumptions on aerosol species or size distribution. Further, the retrieval wavelengths from SAGE III-ISS (521 nm) and CALIPSO (532 nm) are quite close and thus the comparison of the extinction retrievals will not be impacted significantly by errors in the Angstrom exponent. The previous section demonstrated that the retrieved aerosol extinction coefficients reported by the CALIPSO level 3 stratospheric aerosol product agree well with those reported by SAGE III-ISS between 20 km and 30 km within tropical latitudes (30°S-30°N), though the disparities between the two sets of measurements are significantly larger at higher latitudes and lower altitudes. The primary parameter affecting the comparison with SAGE III-ISS is likely to be the lidar ratio used in the CALIPSO retrieval. The CALIPSO extinction retrievals are quite sensitive to the lidar ratio used in the retrieval algorithm (Young et al., 2013, 2016), and the lidar ratio depends upon the optical and physical properties of the scattering particles. In the troposphere, a look-up table of lidar ratios is used by the CALIPSO extinction retrievals for various types of aerosols that might be encountered (Omar et al., 2009, Kim et al, 2018). The version 1.00 level 3 stratospheric aerosol product uses a constant lidar ratio of 50 sr at all latitudes and altitudes for the stratospheric aerosol retrievals. While a lidar ratio of 50 sr has frequently been adopted for stratospheric analyses (e.g. Trickl et al., 2013, Ridley et al., 2014, Sakai et al., 2016, Khaykin et al., 2017), it is not clear if this value is valid all over the stratosphere.

The adopted lidar ratio for the CALIOP stratospheric aerosol retrievals can be assessed by using the independent extinction retrievals from SAGE III-ISS and the attenuated backscatter measurements from CALIOP.  For this we rewrite the Eq. (3c) as

$$T_p^2(z) = \exp\left(-2 \int_0^z \sigma_p(r')dr'\right),$$
(5)

where $\sigma_p$ is the particulate extinction coefficient as retrieved from the occultation measurements from SAGE III-ISS. Using these two-way transmittances from aerosols and computing all other terms in Eq. (2) and Eq. (3) from CALIOP data as earlier, we can obtain an estimate of the particulate backscatter $\beta_p(z)$. The altitude dependent lidar ratio $S_p(z)$ may then be obtained from the expression:

$$S_p(z) = \sigma_p(z) / \beta_p(z)$$
(6)

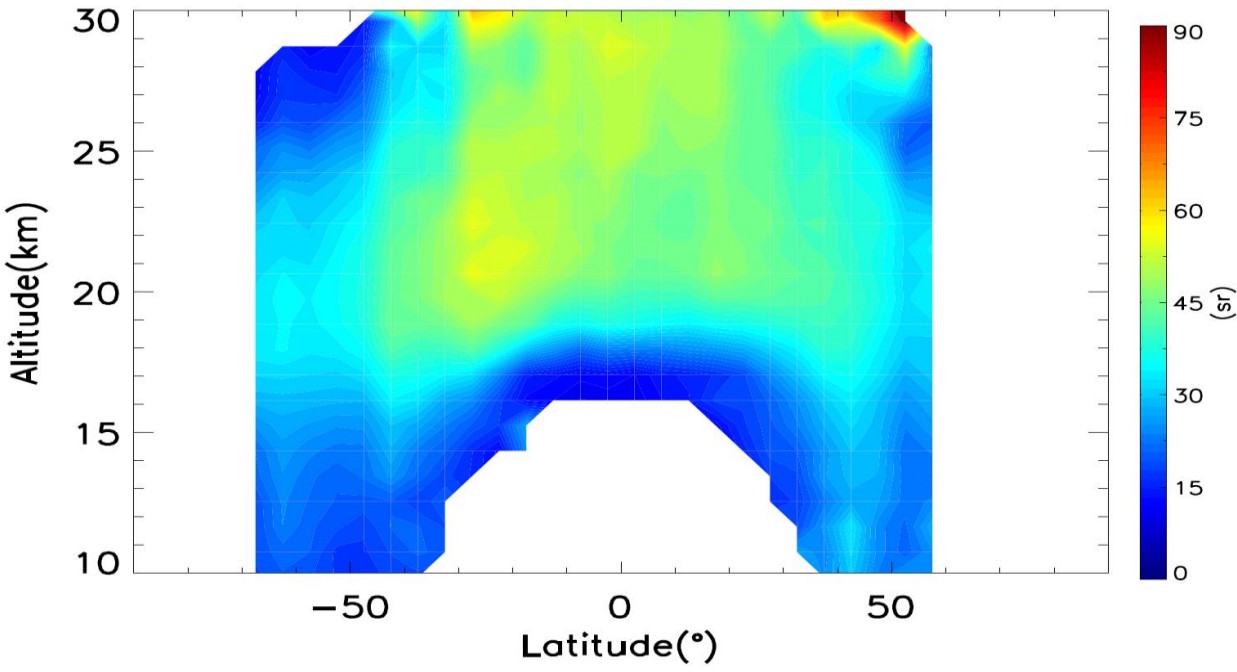

**Figure 14.** Spatial distribution of the stratospheric aerosol 532 nm lidar ratio obtained from the extinction retrievals from SAGE III-ISS and backscatter measurements from CALIOP.

Figure 14 shows the height-latitude cross section of the estimated lidar ratios from the SAGE III-ISS and CALIOP measurements. For this figure, we have used data from both the instruments from June 2017 through March 2018, excluding data from August 2017 through November 2017 to avoid the effect of smoke from the strong pyroCb event of August-September, 2017 as discussed above. The data from both instruments beyond March 2018 were not used to avoid the impact of the Ambae volcano, which erupted in April 2018. As for the comparisons presented in section 3, we have averaged the SAGE III-ISS data over each month and interpolated to the CALIPSO altitude grid and computed the lidar ratios, which were then averaged to obtain the climatological distribution shown in Figure 14. As before, we have cloud cleared the SAGE III-ISS data below 20 km by using only those 521 nm extinction coefficients for which 521 nm to 1022 nm extinction ratio exceeded 2. The Angstrom exponent obtained from these two wavelengths was used to scale the SAGE III-ISS extinction at 521 nm to 532 nm. As can be seen, the lidar ratio values in the bulk of the stratosphere with significant aerosol loading are in the range 45-50 sr, quite similar to the canonical range in the stratosphere (Kremser et al., 2016), with the mean value between 18-30 km and between 40°S and 40°N being $46 \pm 6$ sr. However, in the lowermost stratosphere at all latitudes and in both the polar regions at essentially all altitudes, the estimated lidar ratio values are substantially lower ($\leq 40$ sr, Figure 14). There may be several issues impacting these estimated lidar ratios. We have used the 521 nm aerosol extinction product from SAGE III-ISS, which is still an evolving product. In particular, any errors in the ozone retrievals from SAGE III-ISS are likely to adversely affect the 521 nm aerosol extinction retrievals. Further, in the lowermost stratosphere above the tropopause, mixtures of clouds and aerosols may exist and SAGE III-ISS aerosol data have not been cleared for clouds as such. We have used a simple cloud clearing procedure using the ratio of extinctions at 521 nm and 1022 nm, which might be of limited validity near the tropopause. As mentioned above, incomplete thin cirrus removal artifacts in the CALIPSO stratospheric product, particularly in the "all aerosol" mode may also impact the estimates in the lower stratosphere within a few kilometers above the tropopause particularly in the tropics. Similarly, at high altitudes in the polar regions the aerosol loading is expected to be quite small (extinction $\sim 10^{-5}$ km$^{-1}$) and both sensors are likely to experience difficulty in retrieving these very low extinction coefficients. In particular, CALIOP can have significantly enhanced noise at those altitudes in the polar regions (e.g., see Fig. 16 in Hunt et al., 2009) which may contribute to the differences. Note also that a lidar ratio that is in error at the highest altitudes would lead to incorrect

extinction retrievals for CALIOP lower down, since the attenuation correction that are propagated downward would also be in error. In any case, using a lower lidar ratio in these areas, as suggested by Figure 14, will lead to lower retrieved extinction coefficients from CALIOP and will alleviate the differences noted in section 3.

Is there any evidence of low lidar ratios in the high latitude stratosphere as seen in Figure 14? O'Neill et al. (2012) studied aerosol plumes from the Sarychev volcano from high arctic observations from the Polar Environmental Atmospheric Research Laboratory (PEARL) station (80.05°N, 86.42°W) and estimated a characteristic lidar ratio of 59 sr using AERONET measurements. The Arctic High Spectral Resolution (AHSRL) also retrieved lidar ratio estimates

ranging between 51 sr and 59 sr from measurements acquired between 10 and 15 km. Similarly, Kravitz et al. (2011) reported lidar ratios of 50 sr and 60 sr for Sarychev plumes obtained from the Koldewey Aerosol Raman Lidar (KARL) at Ny- Alesund, Svalbard (78.9°N, 11.9°W). Hoffmann et al. (2010) estimated a somewhat higher value of 65 sr for Kasatochi aerosol plume near the tropopause over Ny-Alesund, in September 2008. Interestingly, and in stark contrast, for a clear

15 day with no Kasatochi layers present, they obtained a background lidar ratio of $18 \pm 6$ sr; however more measurements are clearly needed at all stratospheric altitudes. There is also a paucity of measurements at the stratospheric altitudes over the southern high latitude regions.

         In presence of large ash and even sulfate injections from volcanoes, lidar ratios can be significantly different and can evolve with time. From the so-called "constrained" retrievals, when

the lidar ratios of layers can be obtained directly from the attenuation measurements from CALIOP (Young and Vaughan, 2009), the median lidar ratio of sulfate as well as ash-dominant layers from several volcanoes has been found to be ~ 60-69 sr (Prata et al., 2017, Kar et al., 2018b). In constructing Fig. 14, we have not used data from either instrument for the period of possible contamination from the strong pyroCb event in 2017 as well as after the eruption of Ambae

volcano. We shall continue to refine the lidar ratio estimates using the methodology described here as more contemporaneous measurements from SAGE III-ISS and CALIPSO become available. In future versions of the CALIOP level 3 stratospheric aerosol product we shall attempt to use a more representative lidar ratio over all of the stratosphere.

## 5. Conclusion

In this paper we have provided a detailed account of the algorithm used to construct the recently released CALIPSO level 3 stratospheric aerosol profile product version 1.00. Further, we have given qualitative as well as an initial quantitative assessment of the aerosol extinction retrievals. We have shown that the product captures significant stratospheric aerosol injections (e.g., from volcanic eruptions and wildfires) and clearly illustrates perturbations from stratospheric dynamics over the lifetime of the mission. Comparisons with extinction retrievals obtained from SAGE III-ISSshow quite good agreement to within about 25% in the mean between 20-30 km and between about $30^{o}$S-$30^{o}$N. However, the comparison consistently indicates much larger deviations, exceeding 100-200% (CALIPSO higher), at mid-to-high latitudes ($30^{o}$S-$60^{o}$S and $30^{o}$N-$60^{o}$N) and at low altitudes (10-20 km). The role of the lidar ratio used for the extinction retrievals in the level 3 stratospheric aerosol product was also explored. Based on combined measurements by CALIPSO and SAGE III-ISS, the current lidar ratio of 50 sr is shown to be appropriate for background conditions above 20 km in the tropics. However, it may be unrepresentative of lidar ratios closer to the tropopause and at mid-to-high latitudes. Future versions of the CALIPSO level 3 stratospheric aerosol profile product may refine the lidar ratios based on these and forthcoming analyses.

**Acknowledgements:**

J. K. would like to thank Jean-Paul Vernier, M. C. Pitts and Larry Thomason for useful discussions during the development of the CALIPSO stratospheric product. SAGE III aerosol data were obtained from the Langley Atmospheric Science Data Center. The referees are thanked for useful comments and suggestions that significantly improved the manuscript. J.K. would like to acknowledge illuminating discussions with M. C. Pitts on PSCs.

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
