# Peer review of "CALIPSO Level 3 Stratospheric Aerosol Profile Product: Version 1.00 Algorithm Description and Initial Assessment"

_Atmospheric Measurement Techniques, 2019_

## Referee Comment (RC1) · Andrew Prata (Referee) · 17 Jul 2019

**General comments**

In this contribution, Kar et al. present the new level 3 stratospheric aerosol product for the CALIPSO mission. The details of the science algorithm used to construct the level 3 product are presented. In addition, a preliminary quantitative assessment of the product is made through an inter-comparison of the CALIPSO and SAGE-III (ISS) extinction coefficient retrievals. Some nice observations of volcanic and wildfire smoke aerosols are also described. The paper is well structured and well written and the

assumptions used in the retrieval are clearly articulated. This contribution is important because the level 3 aerosol product could potentially be used in radiative forcing studies that consider the impacts of aerosol loading in the stratosphere. I recommend publication after addressing some minor revisions suggested below.

**Specific comments**

When describing the time-series of stratospheric perturbations due to major volcanic eruptions and wildfires shown in Figure 6, I think it's important to stress in the text (and Abstract) that this analysis is representative of aerosols in the tropical (25°S-25°N) stratosphere. Kasatochi and Sarychev were high latitude eruptions and so most of their sulfates were confined to mid-high latitudes. In addition, Kelud and Nabro are located within tropical latitudes and so their signatures are exaggerated relative to Kasatochi and Sarychev in the figure. Figure 6 would be much more illuminating if panels representing mid-high latitude bands were added.

The discussion on the high bias of the CALIPSO retrievals relative to the SAGE-III retrievals is very interesting. Figure 13 shows that this is largely due to the assumption of a constant lidar ratio set to 50 sr in the CALIPSO product. The authors point out that there were 'probably no significant injections of ash from volcanoes' during their analysis period (June 2017 - August 2018); however, there was a significant ($\sim$0.15 Tg SO2) eruption of Ambae (15.389°S, 167.835°E) in Vanuatu in April 2018 (Global Volcanism Program, 2018). This event may have affected the analysis and should be noted in the discussion section. Another point that could be mentioned is the effect of averaging the data over 15 months. Wouldn't this 'smooth out' the influence of volcanic/smoke aerosols on the derived lidar ratios shown in Figure 13?

Another factor that would impact the new aerosol product is the choice of the color ratio threshold. The authors use a color ratio threshold of 0.5 to remove clouds and retain volcanic ash clouds. However, several authors (Winker et al. 2012; Vernier et

al. 2013; Prata et al., 2017) have shown that volcanic ash colour ratios can be as high as 0.80. Setting this threshold too low may therefore remove volcanic ash from the 'all aerosol' product. This point should be addressed when introducing the choice of their selected threshold.

**Specific comments about figures**

In a lot of the figures the axes and colorbar labels are missing. Also some of the labels are not written clearly. For example, the authors use underscores and abbreviations. I think using proper label names with appropriate variable symbol definitions and units would make the figures clearer. Also latitude/longitude units should use the degree symbol (not the abbreviated 'deg'). At the very least, the labelling should be consistent throughout the paper.

**Technical corrections/suggestions**

P1L28-30: There is also large disagreement ($>$100%) between CALIPSO and SAGE-III at altitudes below 20 km (Figure 11b). This should be stated in the abstract.

P3L26: I see two Kar et al. (2018)s in the references section. Please use 'a' and 'b' to differentiate between them.

P4L10-11: 'The consequences of this change...' - I suggest adding the V3 zonally and vertically averaged attenuated scattering ratio (for the same time period) to Fig. 1. This would make the change from V3 to V4 very clear.

P4L17: Change 'over this latitude' to 'over each latitude'.

P5L17: 'accurate to about 1%' - do you have a reference for this?

P5L28: Delete 'going'.

P6L8: Replace 'i.e.' with 'such as'.

P6L11: 'Vaughan et al. (2009)' - Is there a new reference for the V4 level two layer detection algorithm that you could add?

P6L14: Replace 'but' with 'however'.

P6L18: Change 'product' to 'level 3 stratospheric aerosol product'.

P6L21: Change 'the primary input files used for this product' to 'the primary input file used for the present product'.

P7, Figure 2: For consistency, should use small 'b' in the 'Write results to Background component' box.

P7L7: Please define the 'local tropopause'. E.g. is this taken from GMAO?

P8L4: Change 'Antarctica' to 'Antarctic'. Change 'both the hemispheres' to 'both hemispheres'.

P9L24: Shouldn't this be 'Vernier et al. 2013'?

P9L25: The Puyehue ash did not reach 17 km. Maximum heights observed by CALIOP were ~13 km (Vernier et al., 2013; Prata et al., 2017).

P9L26: Threshold of 0.5 seems too low. Vernier et al. (2013) use a threshold

of 0.8 to discriminate between clouds and volcanic ash. I think the impact of this threshold should be mentioned (see specific comments above).

P10L11: Change 'threshold' to 'threshold of the level 2 layer detection algorithm'.

P11, Figure 4: Can you comment on what's causing the high scattering ratios just above 10 km at ~50°N?

P12L3: What does this look like for a threshold of 0.75-0.80? You may get more of a signal for the Puyehue event.

P12L4: Change 'Nabro' to 'Nabro (near 30°N)'.

P12, Figure 5: I think the labels are wrong here i.e. Figure 5b looks like 'background' and Figure 5a looks like 'all aerosol'.

P12L19-21: I don't see a high number of samples over North America in Figure 5b (see above Figure 5 comment).

P13L27: Change 'significant ash' to 'significant ash and sulfate'.

P14L22: Change 'image' to 'figure'.

P14L23: Change '(Kasatochi, Nabro etc.) to '(e.g. Kasatochi and Nabro)'.

P15L10: 'quite clearly seen' - I'm not sure it is that clear. There are several other features in the figure that are more apparent than the Black Saturday bushfires, which aren't commented on. I suggest changing to 'can be identified'.

P15, Figure 6: I think you could add panels representing middle and high latitude bands to better represent the major stratospheric perturbations on a global scale (see specific comments above). Also, there's a significant feature around December 2010 that's not mentioned. This was probably due to the Merapi (7.54°S, 110.446°E) eruption in Indonesia in November 2010. Surono et al. (2012) estimate 0.44 Tg of SO2 in the upper troposphere and the plume reached heights of 16-17 km. Another feature that's not explained is the one around July 2015. It seems quite significant. Do you know what's causing it?

P16, Figure 7d: Please fix the cropping at the bottom of the figure - some text has been cropped.

P16L15-16: 'irregular shapes' - could this also be due to ice particles?

P17L6: 'smoke spreads globally' - I don't see this in the figure. It looks like the smoke spreads throughout the Northern Hemisphere but the Southern Hemisphere scattering ratio remains unchanged.

P17, Figure 8: What is the cause of the high scattering ratio from 25-30 km over the equator?

P17L17: Kelud erupted in February 2014 not April 2014 (see Kristiansen et al., 2015).

P18L1: 'The gradual lofting of the plume from around 17 km over the tropics to nearly 24 km over several months...'. This seems to imply a rise of 7 km, which I think is misleading. Measuring from the top of the aerosol feature it looks like it rises from 21 to 24 km from March-December 2014 (a rise of 3 km). Please clarify this in the text.

P19L7-9: The Calbuco volcanic cloud actually went almost directly through the SAA (see http://nicarnicaaviation.com/calbuco-eruption-april-2015). Eventually it spread through the Southern Hemisphere but due to the rejection of data in the SAA region a large proportion of the Calbuco signal may not be captured in the CALIPSO level 3 stratospheric aerosol product. I think this is worth mentioning here.

P20L18: Change 'essentially same' to 'essentially the same'.

P20, Equation (4): I got slightly confused here with the notation. What's the difference between $\alpha_p(r)$ (defined at P13L24) and $\sigma(z)_{CALIPSO}$? And which variable is the one that corresponds to the 'all aerosol' profile product?

P22, Figure 11 caption: 'the mean 532 nm extinction coefficient' is this what $\sigma(z)_{CALIPSO}$ is? In Eq. (4) the definition is the 'extinction coefficient at altitude $z$'. I would use the same wording to avoid confusion or put the symbols ($\sigma(z)_{CALIPSO}$ and $\sigma(z)_{SAGE}$) in parentheses in the figure caption.

P22L18: 'the presence of clouds which may impact the retrievals' - Please provide a little more information on how clouds impact the retrieval. If some clouds weren't removed, wouldn't this bias SAGE-III aerosol extinction high? Thus compensating for the difference seen in the comparison with CALIPSO below 20 km?

P22L23: Change '2.0' to '2'.

P22L24: On my first read through, I immediately thought the assumption of constant lidar ratio was the issue. You go on to discuss this but it's not mentioned here. Perhaps it's worth adding a sentence and referencing the discussion that comes later.

P23L15: Change 'Discussion:' to 'Discussion'.

P24L3-5: Change 'tropical latitudes' to 'tropical latitudes (30°S–30°N)'.

P24L5: Change 'higher latitudes' to 'higher latitudes and lower altitudes'.

P24L10: 'smoke, marine aerosols etc' - please list all the aerosol types considered by references cited.

P24, Eq. (5): In Eq. (3), the two-way particulate transmittance is range-dependent. I assume it would be range-dependent ($T_p^2(r)$) here too?

P25L16: 'substantially lower' - Could you put a number to this? E.g. what's the mean lidar ratio in the lowermost stratosphere? I think it is important to give a number or range given the discussion that follows.

P27L3: 'no significant injections of ash from volcanoes' - this is probably true, but there were significant injections of SO2 and therefore sulfate. For example, Ambae (Vanuatu) in April 2018 underwent a significant SO2-rich eruption (see specific comments above).

P27L12: Change 'volcanic eruptions' to 'volcanic eruptions and wildfires'.

P27L16: Change 'mid-to-high latitudes' to 'mid-to-high latitudes (30°S–60°S and 30°N–60°N)'

P27L16: Change 'high altitudes' to 'high altitudes (10–20 km)'

**References**

Global Volcanism Program, 2018. Report on Ambae (Vanuatu). In: Sennert, S K (ed.), Weekly Volcanic Activity Report, 4 April-10 April 2018. Smithsonian Institution and US Geological Survey.

Kristiansen, N. I., Prata, A. J., Stohl, A. and Carn, S. A.: Stratospheric volcanic ash emissions from the 13 February 2014 Kelut eruption, Geophysical Research Letters, 42(2), 588–596, doi:10.1002/2014GL062307, 2015.

Nicarnica Aviation, "Cabulco eruption, April 2015: AIRS Satellite Measurements," 24 April 2015; http://nicarnicaaviation.com/calbuco-eruption-april-2015.

Prata, A. T., Young, S. A., Siems, S. T. and Manton, M. J.: Lidar ratios of stratospheric volcanic ash and sulfate aerosols retrieved from CALIOP measurements, Atmospheric Chemistry and Physics, 17(13), 8599–8618, doi:10.5194/acp-17-8599-2017, 2017.

Surono, Jousset, P., Pallister, J., Boichu, M., Buongiorno, M. F., Budisantoso, A., Costa, F., Andreastuti, S., Prata, F., Schneider, D., Clarisse, L., Humaida, H., Sumarti, S., Bignami, C., Griswold, J., Carn, S., Oppenheimer, C. and Lavigne, F.: The 2010 explosive eruption of Java's Merapi volcano—A '100-year' event, Journal of Volcanology and Geothermal Research, 241–242, 121–135, doi:10.1016/j.jvolgeores.2012.06.018, 2012.

Vernier, J.-P., Fairlie, T. D., Murray, J. J., Tupper, A., Trepte, C., Winker, D., Pelon, J., Garnier, A., Jumelet, J., Pavolonis, M., Omar, A. H. and Powell, K. A.: An Advanced System to Monitor the 3D Structure of Diffuse Volcanic Ash Clouds, Journal of Applied Meteorology and Climatology, 52(9), 2125–2138, doi:10.1175/JAMC-D-12-0279.1, 2013.

Winker, D. M., Liu, Z., Omar, A., Tackett, J. and Fairlie, D.: CALIOP observations of the transport of ash from the Eyjafjallajökull volcano in April 2010, Journal of Geophysical Research: Atmospheres, 117(D20), doi:10.1029/2011JD016499, 2012.

---

## Referee Comment (RC2) · Sergey Khaykin (Referee) · 20 Jul 2019

Please refer to the enclosed pdf

Please also note the supplement to this comment:
https://www.atmos-meas-tech-discuss.net/amt-2019-245/amt-2019-245-RC2-supplement.pdf

---

## Referee Comment (RC3) · Michael Fromm (Referee) · 22 Jul 2019

Hereafter Kar et al. will be shortened to K19. K19 introduce a level 3 stratospheric aerosol product based on CALIOP data. This initial version was developed since the production of the version 4 CALIOP level 1B and 2 data sets. K19 summarize the CALIOP product history and point out the major advance in version 4 that enables a stratospheric L3 product that extends completely through the Junge layer (calibration based on measurements between 36-39 km). They describe how the employ previously documented CALIOP cloud and PSC masks to isolate aerosols, and an additional screen based on a depolarization ratio threshold for creating a separate

background aerosol L3 data set and a background+plume data set. The manuscript is well composed and written. The L3 algorithm is logical and described adequately. K19 offer it as a first version and acknowledge some key areas that may justify refinements. This manuscript is appropriate for AMT. It represents a useful new contribution to the resources that atmospheric and climate scientists need for large-scale studies of the stratosphere. I would recommend publication after K19 satisfactorily address the following issues, one of which I classify as major. Next I characterize this concern, followed by some minor concerns. Finally a list of technical items to address. Major concern: It is apparent from several figures and K19's discussion of them that there may be a significant bias in the L3 products (both background and all-aerosol). My attention was drawn to the residual PSC signature in Figures 4, 8, 9. K19 acknowledge this residual and attribute the signature to "particles in the process of becoming PSC." The residual itself is a dominant feature in all the figure panels where PSC presence is expected. This undoubtedly reveals an incomplete masking of weak but meaningful scattering. In the high-latitude winter realm it is straightforward to dismiss these as tenuous cloud particles, but at lower latitudes and especially the lowermost stratosphere, any residual that may survive cloud screening cannot be simply classified as aerosol. Given that the L3 aerosol algorithm is globally based on the L2 merged layer data, any tenuous cloud (analogous to the acknowledged PSC residual) will be in the L3 dataset and considered aerosol. Just like PSCs, cirrus clouds in their formative and sublimation phase will naturally present a scene that will contain scatterers outside the layer-detection algorithm's thresholds. A ad hoc inspection of L1B quicklooks reveals this to be fairly common. Hence to the extent that a particular scene is subject to thin, patchy, formative or sublimating cirrus, the L3 background and "all aerosol" data sets are at risk for false-positive aerosol detections. This vulnerability reaches its greatest likelihood in regimes of high-frequency, high-altitude cloud. The most prevalent of these is the Asian summer monsoon region, one of the cloudiest on Earth for high cirrus. To the extent that the PSC analogy is accurate for non-polar high-cloud regimes, the L3 background and all-aerosol data may be cloud biased like the PSC signatures

alluded to here. If this concern is well founded, it is essentially impossible to confidently argue that extinction enhancements such as those displayed in Figure 10 in the Asian monsoon sector are aerosols and not tenuous clouds. K19 are encouraged to consider the revelations they showed with respect to PSC residual and assess the applicability of that analogy to the potential for cloud contamination in the lowermost stratosphere, especially where/when cirrus cloud occurrence frequency is high. It might also be instructive to compare Figure 10 with an identical rendering based on the L3 background realization. Ideally the stratospheric background should have no imprint of clouds or aerosol perturbations. The amount of similarity between a background rendering of Figure 10 and the manuscript's rendering would be informative as to how well the L3 algorithm is performing in cirrus-cloudy areas.

This concern was also informed by Figure 3, which shows a layer-like peak in background scattering ratio at the same altitude as the all-aerosol data set in the Sarychev Peak-influenced stratosphere. It led me to wonder what the blue background plot would look like for the same period and geographic cell but for a year without a known volcanic plume. If there is a significant reduction in peak scattering and a monotonic decrease with altitude, it would be suggestive of extra-background aerosols getting into the background data set. Akin to the cloud-detection-threshold conundrum, stratospheric aerosol features absent a feature-detection, yet visible to the eye in quicklook images, are common. To the extent that this is true and quantifiable, L3 background aerosol abundance will be high biased, especially when there are large/widespread tenuous plumes. This may be manifested in Figure 4a, which shows hints of the Nabro plume in the same place as the stronger Nabro feature in the less-screened realizations.

Regarding the depolarization ratio filter adopted by K19, Vernier et al. (2009) ("V09") applied a 5% screen to profiles that are an average of 300-600 L1b profiles. The argumentation therein for 5% was based on an assessment of depol. ratio typical for L1B data. By definition the average depol. ratio in the gridded averaged data is

going to be shifted low with respect to the L1B data. The probability distribution of depol. ratio in the V09 gridded data set is unreported and thus unknown, but it is likely to have a very small mean since it is composed of many clear-sky pixels as well as cloud fringes and weak-cloud pixels. It is unclear how a 5% depol. ratio in such an average distribution maps to 5% in L1B data. Consequently a depol. ratio threshold based on the gridded data may have to be much smaller than even 5%. Neither V09 nor K19 provide any testing in defense of the 5% threshold, hence any conclusions regarding aerosol abundance or cloud contamination in the L3 data set are subject to considerable uncertainty.

An analogous argument applies to the color-ratio filtering described. K19 apply a color-ratio screen based on gridded, averaged data but chose a threshold that is justified in relation to L1B data. Hence some amount of cloud contamination would be systemic in the L3 all-aerosol data set.

K19 are encouraged to assess the veracity of my concern, and if it is valid, to take steps to quantify the biases resulting from inadequate screening.

Minor concerns:

P2, L26-30. This survey of important solar occultation instruments for aerosol measurement is missing a callout for SAM II and POAM II, III. SAM II was especially central to stratospheric aerosol and cloud research. Please consider augmenting this survey.

P5, L23: The L3 data set extends to 85 N and S. This is beyond CALIPSO's orbit extrema of 82 N and S. Please explain the apparent extrapolation.

P8, L07. Two questions regarding the tropopause boundary. Please report the tropopause data that are used. K19 say that tropospheric data are removed. How does that reconcile with Figure 6, which shows an essentially continuous record of tropical aerosol data at 15-16 km? Isn't the tropical tropopause minimally variant and higher than 16 km? Perhaps there are enough tropical profiles where the tropopause

is < 15 km. Some words of clarification are requested.

P11, L10. Regarding the discussion of the June 2011 situation. K19 might consider mentioning Grimsvotn (May 2011, just a bit earlier than June) given that these aerosols are apparently evident in Fig. 4 in the northern extratropics.

P12, discussion of Figure 5. The relative differences are not evident to my eyes. It might be a good idea to show a third panel, A-B. Or perhaps use another color scale to make it more evident.

Figure 5. These sample sizes are huge. Might K19 consider increasing the temporal granularity?

P12, L22. "...particles which are in the process forming PSCs." The same might be said of PSCs in sublimation/evaporation mode. Please consider a minor rewording taking this into account.

P13, L22-30. Discussion of lidar ratio. I did not see any attention to the differences between the lidar ratio for smoke as compared to sulfate. Please augment the discussion in that regard. P14, L22-24. Discussion of Figure 6. This is an unfair comparison between Kelud, a tropical volcano, and a set of extratropical volcanoes. Hence it doesn't seem to be of any value to compare the relative imprints of these plumes. Consider removing this discussion or providing a stronger argument. Figure 6. K19 label and point out some of the obvious features but not all. There seems to be no rationale for this, so please consider labeling the dramatic plumes in J07, J11 (between Sarychev Peak and Nabro), and J16. P15, L09. Discussion of Black Saturday. K19 rightly acknowledge the pyrocumulonimbus source of this stratospheric plume. Dowdy et al., https://doi.org/10.1002/2017JD026577) provide a detailed characterization of the pyroconvection. Please consider citing this paper if K19 consider it important to do so. P15, L09. "...reached altitudes of 16–20 km..." To my eye the plume reached ∼22 km. Am I looking at this correctly? If so, please adjust the description accordingly. P15, L10-11. "These pyrocumulonimbus events seem to be increasing in frequency..." If K19 have

support for that statement please provide it. Perhaps there is a paper to cite?

Figure 7 and discussion thereof. Two points. 1. Please provide a tropopause line or marker. 2. This is a very interesting item to understand but it seems to be out of place with the rest of the paper. Perhaps can offer a strong motivation for including it. If not, consider removing it and making it part of future work.

P16, L13. "...telltale signature of smoke..." the differential 532 nm attenuation is as evident in sulfate plumes as smoke. If this discussion is to remain, it should be expanded to include sulfate.

P18, discussion of Kelud. It should be acknowledged, thanks to CALIPSO, that the injection of Kelud went to 26+km. See Kristiansen et al. (doi:10.1002/2014GL062307) That paper also shows MLS SO2 to 31 mb, so some injected material was up in the mid 20s of km. This means that there may have been no lofting at all, just time-lagged conversion. If K19 agree with this assessment, please consider modifying the discussion accordingly.

P18, L01. "17 km" This is inconsistent with the Fig. First panel shows aerosol to 21 km. Am I looing at this correctly?

Figure 13 and attendant analysis. A few questions and concerns. 1. It wasn't clear to me how K19 matched up the SAGE III data with CALIOP. It's not self evident how that would be done. Please clarify. 2. This figure and text comes in a section called "Discussion" but it is fundamentally a distinct analysis followed by discussion. It would be more logically set off in another titled section. 3. During this analysis timeframe there were background sulfates and fresher smokes from BC2017. The lidar ratios would not be a constant. How have K19 considered this situation?

Technical Matters: In a few places Thomas Trickl's name is misspelled "Trickle." P2, L14: "number" is a singular subject. Replace "number...have" with "number...has" P5, L13: "The V4 data attenuated..." Delete "data." P6, L21: "...the primary input files used for this product is…" Replace "is" with "are." P9, L25: "The distribution…suggest.." Replace "suggest" with "suggests." Figure 4 and other plots with latitude on x-axis: Please explain why the data in summer hemisphere don't extend as far poleward as in the winter hemisphere. P14, L13: "rising trend" This may suggest a nonlinear trend. Perhaps "positive trend" instead? Figure 7. Show the tropopause. Figure 8, bottom panels: What's causing the rainbow edge in the SH? Perhaps trim this off, or explain what is responsible for it. P18, L1: "…lofting…plume from around 17 km…" The figure shows the plume starting out at ∼21 km, not 17 km. Please clarify. P19, L19: Regarding SAGE III ISS providing data "since March 2017…" The SAGE results reported herein start in June 2017. Please clarify.

---

## Referee Comment (RC4) · Anonymous Referee #4 · 7 Aug 2019

The paper "CALIPSO Level 3 Stratospheric Aerosol Product: Version 1.00 Algorithm Description and Initial Assessment" presents and discusses the new science algorithm and data handling techniques that are developed to generate the CALIPSO version 1.00 level 3 stratospheric aerosol profile product. The study falls within the scope of AMT. The authors have done a thorough job, the manuscript is well-written/structured, the presentation clear, the language fluent, the quality of the figures high. The result support the conclusions. Two major deficiencies are the implementation of a constant stratospheric aerosol lidar ratio (50 sr), regardless of an aerosol type classification, and the evaluation of the stratospheric aerosol product against SAGEIII extinction coefficient observations, a product which has not been validated (including issues of

[Figure]

SAGEIII such as cloud contamination propagating in the comparison). However the stratospheric aerosol product and all the issues are properly and extensively discussed in the manuscript, thus I recommend publication in AMT, under minor revisions before it can proceed to be published.

Minor comments:

1) P1L17-18: "gridded level 3 product is based on version 4.2 of the CALIOP level 1 and level 2 data products". According to this sentence CALIOP level 1 V4.2 is used. It is not clear whether the authors refer to Level 1B or Level 1.5 Profile Data. In the case of L1B, please provide a web link to the used data repository. 2) P1L27: "where the average difference between zonal mean extinction profiles is typically less than 25% between 20km and 30km". Please rephrase to provide also whether the sentence refers to overestimation or underestimation compared to SAGEIII. 3) P3L29-30: "This is a level 3 monthly averaged product gridded in latitude (5o), longitude (20o) and altitude (900m)". Although the justification of the 900m vertical resolution is sufficient, the authors should provide explanation on the reasons why the spatial resolution of 5ox20o deg2 grids was selected. How much the selected spatial (horizontal), vertical and temporal resolution affect the final dataset (in terms of backscatter and extinction coefficient profiles at 532nm)? 4) P5L26: "Note that the range of altitudes to be covered in the stratosphere at various latitudes is from 8.2 km to 36 km, the latter being the lower limit of the calibration region". Please mention the applied methodology of decoupling stratospheric and tropospheric layers, since the altitude of 8.2 km frequently lies below Tropopause? Does it rely on MERRA-2 by GMAO? 5) P8L8-9: "Further, all L1B profiles within the South Atlantic Anomaly (SAA) region are also removed": Why do the authors remove CALIOP observations over the SAA region. Based on Kar et al., 2017 (CALIPSO lidar calibration at 532 nm: version 4 nighttime algorithm), the new nighttime CALIOP calibration technique compensates for the higher NSR values, resulting in reliable calibration coefficients even over the SAA region. The authors it is suggested to include the justification in the manuscript. 6) P8L25: "... leading to
generally lower CAD scores (Liu et al., 252019).". Since CAD ranges between -100 and 100, it is not clear whether the authors refer to more aerosol reliable retrievals (CAD -> -100) or to absolute values of CAD score, therefore, CAD values closer to zero. 7) P9L4-7: Although the authors provide the Vaughan et al., 2009 reference, some information on the noise filter should also be included in the manuscript, even if briefly. 8) Figure 4: Based on the manuscript, Figure 4b and 4c refer to the aerosol mode, however it is not clear neither in the caption nor in the manuscript whether they refer to the background or the aerosol mode. In addition, high stratospheric values are observed at 0o latitude, between 25 and 30 km height. Where do the authors attribute the observed values? 9) P12L5: "Note the high scattering ratio values in the Antarctic latitudes between 15 km and 25 km". The authors are kindly requested to provide a reference for this statement. 10) P12L17-18: "The white grid cells over southeast Asia occur because the tropopause is higher than 16 km in this region". The authors are kindly requested to provide a reference for this statement, including the typical tropopause height over this region. 11) P12L21-22: "This is again likely due to small particles which are in the process forming PSCs". The authors are kindly requested to provide a reference for this statement. 12) P13L13: "For the CALIPSO stratospheric aerosol product, the particulate multiple scattering factor is taken as 1 for all species of stratospheric aerosols". The authors are kindly requested to provide a reference for this statement. Which is quantitative the effect of this assumption on the discussed stratospheric aerosol product? 13) P13 - Stratospheric Aerosol Lidar Ratio of 50 sr is used. Although the authors explain in detail the selection of the specific lidar ratio value and evaluate against SAGEIII observations, it is expected that the uniform value used globally, regardless of the aerosol type, introduces large uncertainties. Which is the effect of this assumption to the stratospheric aerosol product? The authors mention that appropriate LR values for different aerosol subtypes will be introduced in future versions of the stratospheric product, however the assumption of constant LR value highly affects the reliability of the extinction coefficient profiles and should be mentioned in the abstract. 14) Figure 8: The authors should discuss on

the high values of attenuated scattering ratios observed over the equator, including the proper references. 15) P18L1-5: "The persistence of the stratospheric perturbation for several months is consistent with the results of Vernier et al. (2016) who found the presence of ash in the lower stratosphere 3 months after the Kelud eruption from balloon observations". The observed features are qualitative consistent with the results of Verner et al. (2016). Is it possible to the authors to include a quantitative comparison? 16) P20L3-5: "SAGE III performs solar and lunar occultation measurements as the ISS orbits the Earth and covers the entire global latitude (90oS to 90oN) and longitude range (180oW to 180oE)." ISS orbital characteristics are characterized by 51.6o inclination, therefore the authors it is suggested to check the global latitude coverage (90oS to 90oN). 17) P20L15-17: "The globally averaged value of the Angstrom exponent derived using all 15 months of data is about 1.56". Please mention between which wavelengths. 18) P20L22: "$\Delta(z) = 100 \times (\sigma(z)CALIPSO - \sigma(z)SAGE)/\sigma(z)SAGE$". How are extreme cases treated? Which computational filters are applied? For instance, cases with $\sigma(z)CALIPSO = 0$ ($\Delta(z) = -1$), or cases with very low values of $\sigma(z)SAGE$ are also included? In case of applied filters in the dataset used prior to the results, the authors should mention them in the manuscript. 19) P22L14: "between CALIPSO and SAGE III extinction at all altitudes with CALIPSO having a high bias". Wherever the manuscript refers to statistical indicators, such as the "high bias" here, the authors should mention the corresponding computed values. 20) P23L8: "calculated using the average extinction coefficient profiles between 20 km and 30 km". The reason of selecting vertically the region between 20km and 30km and not the region from 20 km up to 34 km, hence including the stratospheric region of V3 calibration, is not clear nor justified in the manuscript, since it is proven in Kar et al. (2017) that this region is not aerosol free. 21) P2314: "though the differences begin to rise substantially in the midlatitudes of both hemispheres". Please include explanation on the observed features, including the necessary references.

---

## Author Comment (AC2) · 26 Sep 2019

**Response to Referee#3**

*The article by Jayanta Kar and coauthors presents the new stratospheric aerosol (SA) product based on CALIOP measurements, describes the data handling procedure and provides an initial assessment of the data quality through intercomparison with ISS SAGE III measurements of aerosol extinction. With nearly 12 years of continuous operation, CALIOP measurement record represents a valuable source of near-global information on the stratospheric aerosol variability at seasonal to decadal time scales. An obvious advantage of CALIOP measurements is their higher vertical resolution compared to other space-based aerosol sensors (e.g. SAGE, OSIRIS, OMPS etc.) exploiting passive remote sensing techniques. This is why an official release of CALIOP SA data product has been long awaited by stratospheric community and thus the article represents a valuable contribution. The manuscript is well organized and easy to follow, the data retrieval is comprehensively described and the procedure of cloud screening together with the choice of assumption are well discussed. A novel and valuable result is the latitude-height distribution of extinction to backscatter (lidar) ratio inferred by coupling zonally-averaged CALIOP and SAGE III observations.*

Thanks very much for a careful reading of the manuscript and for your useful suggestions.

*That said, I have a major concern on the data product as such. The very day I found out about the release of L3 CALIOP stratospheric aerosol product, I incorporated the data into the intercomparison of stratospheric AOD series at NH midlatitudes from various satellites and Haute Provence (OHP) lidars in a way we did it in (Khaykin et al., ACP, 2017). The result came quite surprising to me as the new L3 series were remarkably high-biased with respect to all other data sets, including CALIOP SA data product by Vernier et al. I actually thought that I somehow mistreat these data. However, having read this article I realized that this bias is real and amounts to 30-40% at 45 N (Fig. 12), which is consistent with my estimates. A figure below shows time series of AOD of the 17-30 km layer within a 5 deg. latitude belt centered at 44 N as obtained from OHP lidars and various satellite sensors. It includes the CALIOP SA data product by Vernier et al. as well as a more recent one by Friberg et al., (ACP, 2018). While all the data series - independently of the measurement technique (lidar, solar occultation, limb scattering), data handling and the principal measurand (backscatter or extinction) - are in a good agreement, the new L3 CALIOP series stands out high-biased. With that, the background AOD appears low-biased with respect to the well identified clean periods, e.g. 2013-2014.*

[Figure]

We would like to point out two things regarding the intercomparisons plot of AOD you provided. First is the reasonably close match between the CALIPSO L3 product "all aerosol mode" with others for the strong volcanic perturbations. Secondly there is some error in the "background" AOD curve you have plotted. The "background" extinctions do not differ that much from the "all aerosol" mode for quiet conditions. In particular there would be essentially no difference between the two modes above 25 km, since aerosol "layers" hardly ever occur above this altitude and there are no thin cirrus above 25 km. The figure below shows the AODs calculated between 17-30 km from the two modes for the grid cell at 42.5°N:

[Figure]

**General remarks.**
*The article reports the observed bias with respect to SAGE III in an honest and comprehensive way, however the discussion of its possible reasons is not satisfying. Basically, it appeals to inaccurate knowledge of the lidar ratio, cloud screening issues and potential errors in the early version of SAGE III extinction product. However, this can in no way explain the discrepancy with other versions of CALIOP SA products by Vernier et al and Friberg et al., nor with OHP lidar operating at the same wavelength. Obviously, there are other reasons for the positive bias beside the error in lidar ratio or that of SAGE III extinction product. These reasons are neither identified nor hypothesized upon, leaving one wonder about the credibility of the L3 SA product as a whole and strongly limiting its scientific value, particularly for radiative forcing studies. Another missing item is the discussion on the quality of the "background aerosol" product. I suggest that the authors attempt to investigate the possible reasons for the latitude and altitude dependent bias and try to eliminate it if possible or at least sketch the envisaged changes/improvements in the future version of CALIOP L3 SA product, other than refinement of lidar ratios. In order to isolate the lidar ratio issue, the validation of the L3 data product could be done on integrated backscatter available from NDACC ground-based lidars at various latitudes.*

As described in the text, we rigorously solve the lidar equation from 36 km downwards to 1 km below the tropopause using the same extinction retrieval module as for the standard CALIPSO tropospheric retrieval. In contrast, the Vernier et al. product does not solve the lidar equation, but instead compute the extinction by multiplying the gridded attenuated backscatter  by a lidar ratio

of 50 sr. As a result, they do not correct the attenuated backscatter for the particulate attenuation at the successive range bins. This may not make significant difference in the retrieved extinction above 20 km under low aerosol loading. However the attenuation can quickly build up below 20 km and under high aerosol loading leading to a significant low bias in the Vernier et al. extinction profiles. Friberg et al. (2018) recognized this and pointed out the large underestimation in AOD that can result from this effect, particularly in the lowermost stratosphere. Friberg et al.(2018) did apply an approximate particulate attenuation correction to their product, nonetheless, even they did not rigorously solve the lidar equation.

In the manuscript, we have pointed out the high bias in our extinction values in mid-high latitude. Some of this can come from cloud clearance issues, either from the inadequacies of our CAD algorithm or from difficulties in complete removal of thin cirrus clouds, the latter mostly affecting a few kilometers above the tropopause in the tropics. We have now added a new figure (Figure 4) and discussed possible cloud contamination issues in both the modes. However, what needs to be appreciated is the retrieval of extremely low values of extinction ($\sim 10^{-4}$ - $10^{-5}$ km$^{-1}$) that is being attempted using the CALIOP backscatter measurements with significantly lower SNR than in the troposphere. We believe we have discussed the possible issues impacting the differences with SAGE III to the best of our knowledge at this time. However, we do believe that the more significant contributor to this bias at mid-high latitudes is the lidar ratio, as also becomes quite clear from Figure 13 (now Figure 14) in the manuscript. In any case we shall continue to investigate the reasons for these biases further using stratospheric extinction retrievals from other satellites and NDACC ground based lidars at various locations (as you have suggested). However that is beyond the scope of this manuscript; here we are primarily describing the algorithm involved in retrieving the extinction profiles.

**Specific remarks**

*Figure 4. A strong signal above southern high latitudes is certainly due to PSC and I believe these are type Ib PSC (supercooled ternary solution, STS), which are non depolarizing and thus may be aliased as stratospheric aerosol. The interpretation in p.12 l.5-7 (signature of particles in the process of becoming PSCS) is thus incorrect. I wonder, why the PSCs could not be screened out using temperature threshold for PSC formation, which are relatively well known.*

As mentioned in the text, we have used the level 1 CALIOP data for this product while using the level 2 products to filter out the cloud layers. Globally we have used the level 2 clouds as detected by our CAD algorithm and for consistency we decided to use the level 2 CALIPSO PSC mask product to remove the PSCs. Clearly the backscatter is from residual material below the detection limit of the PSC algorithm and likely to be the background aerosol particles which are transitioning into full blown detectable PSC layers. There is only a fine line between background stratospheric aerosol and when it starts growing by HNO$_3$ uptake and thus may be considered a PSC particle. We have modified the sentence in line 5, page 12 as:

"Since all PSC layers detected by the dedicated PSC detection algorithm were removed, what remains are the signatures of only those particles below the detectability threshold of the PSC mask data product."

Also, using the temperatures from MERRA-2 at these high latitudes as a threshold for detecting PSCs may have its own caveats. For this first version of the product, we have not taken that approach.

***Figure 8 and 9, both showing latitude-height sections. Is there a particular reason why the former reports the attenuated scattering ratio, whereas the latter reports extinction coefficient? It would be easier to compare them had they presented the same units.***

The attenuated scattering ratio and the retrieved extinction coefficient are both part of the product and we wanted to show examples from both the variables.

***Figure 9. What causes the strong signal around the tropopause at midlatitudes? If this is cloud contamination, this should be carefully discussed.***

We have added the following:

"Note the high extinction values near 50ºN-60ºN in the lower stratosphere (~10-15km). These are similar to the summer rise in extinctions at these latitudes as discussed earlier (Figure 7) and are possibly due to biomass burning effects but could also be related to possible cloud clearance issues. As also mentioned above, the high extinctions at high southern latitudes could be related to scattering from particles below the PSC detection threshold as well as to transported volcanic material from Kelud."

---

## Author Comment (AC3) · 26 Sep 2019

**Response to Referee #1**

*Hereafter Kar et al. will be shortened to K19. K19 introduce a level 3 stratospheric aerosol product based on CALIOP data. This initial version was developed since the production of the version 4 CALIOP level 1B and 2 data sets. K19 summarize the CALIOP product history and point out the major advance in version 4 that enables a stratospheric L3 product that extends completely through the Junge layer (calibration based on measurements between 36-39 km). They describe how the employ previously documented CALIOP cloud and PSC masks to isolate aerosols, and an additional screen based on a depolarization ratio threshold for creating a separate background aerosol L3 data set and a background+plume data set. The manuscript is well composed and written. The L3 algorithm is logical and described adequately. K19 offer it as a first version and acknowledge some key areas that may justify refinements. This manuscript is appropriate for AMT. It represents a useful new contribution to the resources that atmospheric and climate scientists need for large-scale studies of the stratosphere. I would recommend publication after K19 satisfactorily address the following issues, one of which I classify as major. Next I characterize this concern, followed by some minor concerns. Finally a list of technical items to address.*

Thanks very much for a careful reading of the manuscript and for your useful suggestions.

*Major concern: It is apparent from several figures and K19's discussion of them that there may be a significant bias in the L3 products (both background and all-aerosol). My attention was drawn to the residual PSC signature in Figures 4, 8, 9. K19 acknowledge this residual and attribute the signature to "particles in the process of becoming PSC." The residual itself is a dominant feature in all the figure panels where PSC presence is expected. This undoubtedly reveals an incomplete masking of weak but meaningful scattering. In the high-latitude winter realm it is straightforward to dismiss these as tenuous cloud particles, but at lower latitudes and especially the lowermost stratosphere, any residual that may survive cloud screening cannot be simply classified as aerosol. Given that the L3 aerosol algorithm is globally based on the L2 merged layer data, any tenuous cloud (analogous to the acknowledged PSC residual) will be in the L3 dataset and considered aerosol. Just like PSCs, cirrus clouds in their formative and sublimation phase will naturally present a scene that will contain scatterers outside the layer-detection algorithm's thresholds. An ad hoc inspection of L1B quicklooks reveals this to be fairly common. Hence to the extent that a particular scene is subject to thin, patchy, formative or sublimating cirrus, the L3 background and "all aerosol" data sets are at risk for false-positive aerosol detections. This vulnerability reaches its greatest likelihood in regimes of high-frequency, high-altitude cloud. The most prevalent of these is the Asian summer monsoon region, one of the cloudiest on Earth for high cirrus. To the extent that the PSC analogy is accurate for non-polar high-cloud regimes, the L3 background and all-aerosol data may be cloud biased like the PSC signatures alluded to here. If this concern is well founded, it is essentially impossible to confidently argue that extinction enhancements such as those displayed in Figure 10 in the Asian monsoon sector are aerosols and not tenuous clouds. K19 are encouraged to consider the revelations they showed with respect to PSC residual and assess the applicability of that analogy to the potential for cloud contamination in the lowermost stratosphere, especially where/when cirrus cloud occurrence frequency is high. It might also be instructive to compare Figure 10 with an identical rendering based on the L3 background realization. Ideally the stratospheric background should have no imprint of clouds or aerosol perturbations. The amount of similarity between a background rendering of Figure 10 and the manuscript's rendering would be informative as to how well the L3 algorithm is performing in cirrus-cloudy areas.*

*This concern was also informed by Figure 3, which shows a layer-like peak in background scattering ratio at the same altitude as the all-aerosol data set in the Sarychev Peak-influenced stratosphere. It led me to wonder what the blue background plot would look like for the same period and geographic cell but for a year without a known volcanic plume. If there is a significant reduction in peak scattering and a monotonic decrease with altitude, it would be suggestive of extra-background aerosols getting into the background data set. Akin to the cloud-detection-threshold conundrum, stratospheric aerosol features absent a feature-detection, yet visible to the eye in quicklook images, are common. To the extent that this is true and quantifiable, L3 background aerosol abundance will be high biased, especially when there are large/widespread tenuous plumes. This may be manifested in Figure 4a, which shows hints of the Nabro plume in the same place as the stronger Nabro feature in the less-screened realizations.*

Your concern about cloud contamination is a perennial problem with all satellite instruments attempting to retrieve aerosol in the lower stratosphere below about 20 km. We tried our best to remove cloud contamination by using several different filters. As mentioned in the manuscript, the first filter removes the layers as detected by the CALIPSO layer detection module and then classified as "cloud"s by the Cloud-Aerosol-Discrimination (CAD) module. Clearly, the efficiency of removal of cloud layers in this dataset depends upon the efficiency of both these algorithms. In particular the CAD algorithm uses the optical properties of the layers, specifically the volume depolarization ratio and attenuated color ratio with both measurements becoming noisy at increasing altitudes above the local tropopause. The second filter removes the PSCs as identified by the separate PSC Mask product and again depends upon the robustness of the PSC product. Note that we have used v1.0 of the PSC product which has since been updated into V1.5 and we will use the latter in the future versions of the stratospheric aerosol product. For this version of the product and for the sake of uniformity, we decided to use the clouds (PSCs or otherwise) as detected by the available CALIPSO layer products. Note that in the polar regions, particularly over Antarctica, layers detected as "clouds" by the CAD algorithm and independently as PSCs have been removed. As has been clearly stated in the manuscript, scattering from particles below the threshold of detection for both the regular layers as well as the PSC layers will remain in the product. In particular, in winter over Antarctica, it may not be easy to define the point when the background aerosol transitions into a detectable PSC.

As you mentioned and as is well-known, it is very difficult to remove the tenuous cirrus clouds which occur within a few kilometers near the tropical tropopause. Our third filter attempts to do this using a threshold on either volume depolarization ratio or attenuated color ratio. We discuss these filters further below. However, we would like to point out that presence of aerosols within the Asian summer monsoon anticyclone between ~13-18 km (the so-called Asian Tropopause Aerosol Layer, ATAL), formed from gas phase precursors or primary aerosols related to deep convection has been confirmed from balloon and aircraft observations, although some ice clouds are likely present over this same area (Tobo et al., 2007, Vernier et al., 2011, 2015, 2018, Hopfner et al., 2019). Therefore most of this plume as e.g. seen in Fig 10 (Fig. 11 in revised version) is likely aerosol.

[Figure]

The figure above shows the spatial distribution of the "background" aerosol mode for July 2015 at 17 km. As can be seen, the Asian summer monsoon plume as well as the signature of the Calbuco volcano between 30°S and 50°S appear even in the background mode. As discussed above, any signal that is below the layer detection threshold and survives the different filters mentioned above will be included in our background product.

*Regarding the depolarization ratio filter adopted by K19, Vernier et al. (2009) ("V09") applied a 5% screen to profiles that are an average of 300-600 L1b profiles. The argumentation therein for 5% was based on an assessment of depol. ratio typical for L1B data. By definition the average depol. ratio in the gridded averaged data is going to be shifted low with respect to the L1B data. The probability distribution of depol. ratio in the V09 gridded data set is unreported and thus unknown, but it is likely to have a very small mean since it is composed of many clear-sky pixels as well as cloud fringes and weak-cloud pixels. It is unclear how a 5% depol. ratio in such an average distribution maps to 5% in L1B data. Consequently a depol. ratio threshold based on the gridded data may have to be much smaller than even 5%. Neither V09 nor K19 provide any testing in defense of the 5% threshold, hence any conclusions regarding aerosol abundance or cloud contamination in the L3 data set are subject to considerable uncertainty. An analogous argument applies to the color-ratio filtering described. K19 apply a color ratio screen based on gridded, averaged data but chose a threshold that is justified in relation to L1B data. Hence some amount of cloud contamination would be systemic in the L3 all-aerosol data set. K19 are encouraged to assess the veracity of my concern, and if it is valid, to take steps to quantify the biases resulting from inadequate screening.*

As mentioned above and stated in the manuscript, the depolarization and the color ratio filters were primarily designed to remove the thin cirrus near the tropopause. This is a non-trivial issue if at the same time you are trying to capture the signals of volcanic ash in the stratosphere in the early part of the plume evolution which is important to characterize from the point of view of aviation safety. Firstly the aerosol/cloud discrimination depends upon the scattering ratio or the strength of the plume, and in particular there is generally a large overlap between aerosols and clouds at low scattering ratios. Clearly no single number can be optimal for all situations or at all scattering ratios. We used the cirrus filter at the gridded level to avoid the noise at the profile-by-profile basis. We used June 2011 for testing the filters because of the two representative cases, Nabro being mostly sulfate and Puyehue-Cordon Caulle (PCC) being mostly ash and we wanted to capture both of these stratospheric perturbations. In particular it was a good test case for capturing ash plumes from PCC, since the ash plumes were observed ~12 km, near the cruising altitudes of most airline flights. As explained in the text it was not possible to achieve this using only a depolarization filter to remove the cirrus clouds—indeed the volume depolarization ratio of optically thin cirrus clouds can be as low as 0.1, so even using a very low volume depolarization ratio it would not be possible to get rid of all cirrus.  On the other hand, as shown in Figure 4 ( Figure 3 in revised version), the attenuated color ratio filter with threshold at 0.5 at the gridded level (which may include both high and low scattering ratios),  did capture the ash plumes of PCC also retaining the Nabro plumes at the same time. Lowering the threshold to, say 0.1 for the attenuated color ratio, signals from both the volcanoes essentially disappear and using a higher threshold (e.g. 0.8 which might be more appropriate for higher scattering ratios as also pointed out by referee#2) may retain much more clouds. However, for tenuous plumes, the adopted threshold of 0.5 is likely to retain some cirrus cloud at the tropics near the tropopause, which cannot be helped since we use the same threshold uniformly for all cases. We have now explained this in the text using a new Figure (Figure 4). As shown in that Figure, for such tenuous plumes, the aerosol extinction does not change significantly between the "all aerosol" and "background" modes, but the depolarization filter in the background mode removes the cirrus more efficiently. Thus it might be better to use the "background" mode for volcanically quiet years or for sulfate volcanoes, particularly after several weeks when the

plumes become tenuous. In any case, as seen in the climatology of thin cirrus (Sassen et al. 2008), this will primarily impact only the tropics and in the vicinity of the tropopause (up to ~17-18km). In particular, any bias seen in comparisons with other datasets above ~ 20 km will have nothing to do with this cirrus filter and there may be other sources for such a bias.

*Minor concerns:*
*P2, L26-30. This survey of important solar occultation instruments for aerosol measurement is missing a callout for SAM II and POAM II, III. SAM II was especially central to stratospheric aerosol and cloud research. Please consider augmenting this survey.*

We have revised the relevant paragraph in the revised version as:

"Most of our current knowledge of the global distribution of stratospheric aerosols comes from satellite measurements. The earliest such measurements were carried out by the Stratospheric Aerosol Measurement II (SAM II) on board the Nimbus 7 spacecraft which provided the vertical profiles of aerosol extinction at 1 μm and were followed by the Stratospheric Aerosol and Gas Experiment (SAGE) series of instruments (Chu and McCormick, 1979, Kent and McCormick, 1984, Mauldin et al., 1985; Chu et al., 1993; Damadeo et al., 2013). The basic principle employed in these instruments is solar occultation, wherein the vertical profile of stratospheric aerosols is retrieved from measurement of sunlight as the rays pass through the atmosphere during sunrise and sunset events as observed from the orbiting spacecraft. Stratospheric aerosols have been characterized using this technique from SAGE instruments on Earth Radiation Budget Satellite (ERBS) and Meteor-3M as well as from the International Space Station (ISS). Among other space-borne instruments that use this technique are the Polar Ozone and Aerosol Measurement (POAM II, POAM III, Glaccum et al., 1996, Lucke et al., 1999) and Measurement of Aerosol Extinction in the Stratosphere and Troposphere Retrieved by Occultation (MAESTRO, McElroy et al., 2009). In addition, the Optical Spectrograph and InfraRed Imager System (OSIRIS) and the Ozone Mapping and Profiler Suite (OMPS) have used a limb scatter technique to obtain the aerosol extinction profiles (Bourassa et al., 2012, Chen et al., 2018)."

*P5, L23: The L3 data set extends to 85 N and S. This is beyond CALIPSO's orbit extrema of 82 N and S. Please explain the apparent extrapolation.*

Since the data are binned in 5° in latitude, the first and the last bins imply available data averaged between 85° and 80° in both hemispheres.

*P8, L07. Two questions regarding the tropopause boundary. Please report the tropopause data that are used. K19 say that tropospheric data are removed. How does that reconcile with Figure 6, which shows*

*an essentially continuous record of tropical aerosol data at 15-16 km? Isn't the tropical tropopause minimally variant and higher than 16 km? Perhaps there are enough tropical profiles where the tropopause is < 15 km. Some words of clarification are requested.*

We have added the following:

"The tropopause heights were taken from the Modern-Era Retrospective analysis for Research and Applications 2 (MERRA-2) reanalyses as in all V4 products (Gelaro et al., 2017)."

Please note that Figure 6 has been revised (now Figure 7) and two new panels have been added showing the distribution at mid-high latitudes in both hemispheres. Also, the stratospheric retrievals for this first version were carried out up to one km below the local tropopause for use in UTLS studies which would explain the extinction data below 16 km. In any case the tropical distribution is now shown starting at 17 km.

*P11, L10. Regarding the discussion of the June 2011 situation. K19 might consider mentioning Grimsvotn (May 2011, just a bit earlier than June) given that these aerosols are apparently evident in Fig. 4 in the northern extratropics.*

Thanks for this information. We have added the following:

"Further, the high scattering ratios near 50ºN are likely due to the Grimsvotn volcano, which erupted in May 2011."

*P12, discussion of Figure 5. The relative differences are not evident to my eyes. It might be a good idea to show a third panel, A-B. Or perhaps use another color scale to make it more evident.*

*Figure 5. These sample sizes are huge. Might K19 consider increasing the temporal granularity?*

Per your suggestion, we have now added a third panel showing the difference in samples between the "all aerosol" and "background" modes.

We are not exactly sure what you mean by "temporal granularity". Figure 6 (previously Figure 5) shows the number of samples accumulated at an altitude of 17 km in each horizontal grid cell for the month of July 2009. Since the level 3 stratospheric aerosol data product reports monthly averages aggregated on $5°$ latitude $\times$ $20°$ longitude spatial grid, the "temporal granularity" of the product is one month.

*P12, L22. ". . .particles which are in the process forming PSCs." The same might be said of PSCs in sublimation/evaporation mode. Please consider a minor rewording taking this into account.*

We have deleted this sentence and revised the previous sentence as:

"Also note the high number of samples over parts of Antarctica, partly from oversampling due to orbital configuration and related to small particles below the detectability of PSCs by the PSC mask algorithm."

*P13, L22-30. Discussion of lidar ratio. I did not see any attention to the differences between the lidar ratio for smoke as compared to sulfate. Please augment the discussion in that regard.*

We have added the following in the text:

"Note that the lidar ratios could also be significantly different for stratospheric perturbations resulting from smoke intrusion from pyroCb events (Peterson et al., 2018, Khaykin et al., 2018). For this first version of CALIPSO stratospheric aerosol product we have used only a single lidar ratio."

*Discussion of Figure 6. This is an unfair comparison between Kelud, a tropical volcano, and a set of extratropical volcanoes. Hence it doesn't seem to be of any value to compare the relative imprints of these plumes. Consider removing this discussion or providing a stronger argument.*

We have deleted the relevant sentence. As already mentioned above, we have now added a couple of new panels (new Figure 7) showing the time altitude plots at mid/high latitudes in both hemispheres, per the second referee's suggestion. The discussion has been modified accordingly.

*Figure 6. K19 label and point out some of the obvious features but not all. There seems to be no rationale for this, so please consider labeling the dramatic plumes in J07, J11 (between Sarychev Peak and Nabro), and J16.*

We wanted to point out the outstanding features without cluttering the plot with too many labels. In any case, we have now added labels for the features you have mentioned.

*P15, L09. Discussion of Black Saturday. K19 rightly acknowledge the pyrocumulonimbus source of this stratospheric plume. Dowdy et al., https://doi.org/10.1002/2017JD026577) provide a detailed characterization of the pyroconvection. Please consider citing this paper if K19 consider it important to do so.*

Dowdy et al. mostly deal with lightnings associated with the Black Saturday event as well as fires ignited by the event and don't quite directly address the stratospheric aerosol plumes being discussed in Figure 6 (now Figure 7) and so we have not cited this paper.

*P15, L09. ". . .reached altitudes of 16–20 km. . ." To my eye the plume reached _22 km. Am I looking at this correctly? If so, please adjust the description accordingly.*

That sentence referred to the findings from the cited works. We have added the following sentence: "The signature of this event can also be identified in Figure 7b, reaching up to nearly 22 km".

**P15, L10-11. "These pyrocumulonimbus events seem to be increasing in frequency. . ." If K19 have support for that statement please provide it. Perhaps there is a paper to cite?**

We have deleted this sentence in the revised version.

*Figure 7 and discussion thereof. Two points. 1. Please provide a tropopause line or marker. 2. This is a very interesting item to understand but it seems to be out of place with the rest of the paper. Perhaps can offer a strong motivation for including it. If not, consider removing it and making it part of future work.*

*P16, L13. ". . .telltale signature of smoke. . ." the differential 532 nm attenuation is as evident in sulfate plumes as smoke. If this discussion is to remain, it should be expanded to include sulfate.*

Per your suggestion, we have now added the tropopause lines in all the four panels in this figure (now Figure 8). We believe it is a relevant figure in that it sets the stage for the height-latitude cross sections of scattering ratios we show in the following figure. We can designate the encircled plumes in Figure 9 as smoke only on the basis of the discussion of this figure.

The level 2 stratospheric aerosol classification algorithm classifies most of the layers as "smoke" or "ash". As mentioned in the manuscript, some of the "ash" has been misclassified because of high depolarization. However we did not mention "sulfate" anywhere.

***P18, discussion of Kelud. It should be acknowledged, thanks to CALIPSO, that the injection of Kelud went to 26+km. See Kristiansen et al. (doi:10.1002/2014GL062307). That paper also shows MLS SO2 to 31 mb, so some injected material was up in the mid 20s of km. This means that there may have been no lofting at all, just time-lagged conversion. If K19 agree with this assessment, please consider modifying the discussion accordingly.***

Kristiansen et al. point out that most of the ash injection was around 17 km. Also from our Figure 9 (Figure 10 in revised version), the slow ascent of the plume can be clearly seen which is consistent with the upward branch of the Brewer-Dobson circulation. Friberg et al. (2018) also support this explanation. Therefore we have not modified the discussion as such.

**P18, L01. "17 km" This is inconsistent with the Fig. First panel shows aerosol to 21 km. Am I looking at this correctly?**

We have modified the sentence as follows:

"The gradual lofting of the plume, with its top rising from ~21 km over the tropics in March to ~24 km in the same general location several months later, shows the signature of stratospheric dynamics in the CALIPSO stratospheric aerosol product."

***Figure 13 and attendant analysis. A few questions and concerns. 1. It wasn't clear to me how K19 matched up the SAGE III data with CALIOP. It's not self evident how that would be done. Please clarify. 2. This figure and text comes in a section called "Discussion" but it is fundamentally a distinct analysis followed by discussion. It would be more logically set off in another titled section. 3. During this analysis timeframe there were background sulfates and fresher smokes from BC2017. The lidar ratios would not be a constant. How have K19 considered this situation?***

This figure is the first attempt to obtain the stratospheric lidar ratio using backscatter measurements from CALIPSO and extinction data from SAGE III, using the limited amount of contemporaneous data available at this time. Hence we have done it only in a climatological sense. In other words, we have not used strictly coincident profiles from each instrument as such. We have added the following to clarify this:

"As for the comparisons presented in section 3, we have averaged the SAGE III data over each month and interpolated to the CALIPSO altitude grid and computed the lidar ratios, which were then averaged to obtain the climatological distribution shown in Figure 14."

Even though it is a new result, it fits in better in the context of the discussion on the lidar ratios. So we have decided to let this be a part of the discussion itself.

Please note that we have now replotted this figure by not using data from the biomass burning months of August through November 2017 and also all data after March 2018 to avoid

contamination from the Ambae volcano. As can be seen in the revised figure, the results remain much the same despite loss of data.

*Technical Matters:*

*In a few places Thomas Trickl's name is misspelled "Trickle."*

*P2, L14: "number" is a singular subject.*

*P5, L13: "The V4 data attenuated. . ." Delete "data."*

*P6, L21: ". . .the primary input files used for this product is. . ." Replace "is" with "are."*

*P9, L25: "The distribution. . .suggest.." Replace "suggest" with "suggests."*

These points have been taken care of.

*Figure 4 and other plots with latitude on x-axis: Please explain why the data in summer hemisphere don't extend as far poleward as in the winter hemisphere.*

Added a sentence in the legend clarifying lack of data at high northern latitudes during nighttime in summer months.

*P14, L13: "rising trend" This may suggest a nonlinear trend. Perhaps "positive trend" instead?*

Done.

*Figure 7. Show the tropopause.*

Done.

*Figure 8, bottom panels: What's causing the rainbow edge in the SH? Perhaps trim this off, or explain what is responsible for it.*

Done.

*P18, L1: ". . .lofting. . .plume from around 17 km. . ." The figure shows the plume starting out at _21 km, not 17 km. Please clarify.*

We have revised the text as:

"The gradual lofting of the plume, with its top rising from ~21 km over the tropics in March to ~24 km in the same general location several months later, shows the signature of stratospheric dynamics in the CALIPSO stratospheric aerosol product.."

*P19, L19: Regarding SAGE III ISS providing data "since March 2017. . ." The SAGE results reported herein start in June 2017. Please clarify.*

Although the ASDC archives state the temporal coverage for the SAGE III aerosol data start from March 2017, the available data actually begins in June 2017 after commissioning. We have modified the sentence as: "The aerosol extinction profiles are available from the solar occultation measurements in 9 channels from 384 nm to 1544 nm starting June 2017."

**References**:

Friberg, J., et al., Atmos. Chem. Phys., 18, 11149, https://doi.org/10.5194/acp-18-11149-2018., 2018.

Hopfner, M., et al., Nature Geosc., https://doi.org/10.1038/s41561-019-0385-8, 2019.

Tobo, Y et al., Atmos. Res., 84, 233-241, https://doi.org/10.1016/j.atmosres. 2006.08.003, 2007.

Vernier, J.-P. et al., Geophys. Res. Lett., 38, L07804, https://doi.org/10.1029/2010GL046614, 2011.

Vernier , J.-P, et al., J. Geophys. Res. Atmos., 120, 1608, doi:10.1002/2014JD022372, 2015.

Vernier,J.-P.,et al., Bull. Am. Meteorol.Soc. **99**, 955–973, https://doi.org/10.1175/BAMS-D-17-0014.1, 2018.

---

## Author Comment (AC1)

**Response to Referee #2**

**General comments**

In this contribution, Kar et al. present the new level 3 stratospheric aerosol product for the CALIPSO mission. The details of the science algorithm used to construct the level 3 product are presented. In addition, a preliminary quantitative assessment of the product is made through an inter-comparison of the CALIPSO and SAGE-III (ISS) extinction coefficient retrievals. Some nice observations of volcanic and wildfire smoke aerosols are also described. The paper is well structured and well written and the assumptions used in the retrieval are clearly articulated. This contribution is important because the level 3 aerosol product could potentially be used in radiative forcing studies that consider the impacts of aerosol loading in the stratosphere. I recommend publication after addressing some minor revisions suggested below.

Thanks very much for a careful reading of the manuscript and for your useful suggestions.

**Specific comments**

When describing the time-series of stratospheric perturbations due to major volcanic eruptions and wildfires shown in Figure 6, I think it's important to stress in the text (and Abstract) that this analysis is representative of aerosols in the tropical (25\_S-25\_N) stratosphere. Kasatochi and Sarychev were high latitude eruptions and so most of their sulfates were confined to mid-high latitudes. In addition, Kelud and Nabro are located within tropical latitudes and so their signatures are exaggerated relative to Kasatochi and Sarychev in the figure. Figure 6 would be much more illuminating if panels representing mid-high latitude bands were added.

Per your suggestion, we have revised this Figure (now Figure 7) with two new panels on mid-high southern (40°S-60°S) and northern latitudes (40°N-60°N). We have also revised the relevant sentence in the abstract as:

"Further, we show that the extinction profiles (retrieved using a constant lidar ratio of 50 sr) capture the major stratospheric perturbations in both hemispheres over the last decade resulting from volcanic eruptions, extreme smoke events, and signatures of stratospheric dynamics."

The discussion on the high bias of the CALIPSO retrievals relative to the SAGE-III retrievals is very interesting. Figure 13 shows that this is largely due to the assumption of a constant lidar ratio set to 50 sr in the CALIPSO product. The authors point out that there were 'probably no significant injections of ash from volcanoes' during their analysis period (June 2017 - August 2018); however, there was a significant (\_0.15Tg SO2) eruption of Ambae (15.389\_S, 167.835\_E) in Vanuatu in April 2018 (Global Volcanism Program, 2018). This event may have affected the analysis and should be noted in the discussion section. Another point that could be mentioned is the effect of averaging the data over 15 months. Wouldn't this 'smooth out' the influence of volcanic/smoke aerosols on the derived lidar ratios shown in Figure 13?

Thank you for pointing this out. We have now replotted this figure (now Figure 14) by not using data from the biomass burning months of August through November 2017 and also all data after March 2018 to avoid contamination from the Ambae volcano. As can be seen in the revised figure, the results remain much the same despite loss of data.

Another factor that would impact the new aerosol product is the choice of the color ratio threshold. The authors use a color ratio threshold of 0.5 to remove clouds and retain volcanic ash clouds. However, several authors (Winker et al. 2012; Vernier et al. 2013; Prata et al., 2017) have shown that volcanic ash colour ratios can be as high as 0.80. Setting this threshold too low may therefore remove volcanic ash

**from the 'all aerosol' product. This point should be addressed when introducing the choice of their selected threshold.**

We have discussed this issue in detail in response to the first referee's comments. Essentially the problem stems from using one single hard number as threshold for all situations, which is what we have done for this first version of the stratospheric data product. Pueyhue Cordon Caulle (PCC) volcano had a large number of plumes with strong scattering. So using a larger threshold for the attenuated color ratio like 0.8 will capture more of those plumes. However, the larger particles with such high attenuated color ratios will sediment out relatively rapidly leaving more diffuse ash with low scattering ratios. As mentioned in the response to the first referee, at low scattering ratios there can be a significant overlap between the thin cirrus and aerosols. Using a attenuated color ratio threshold of 0.8 will thus include a larger contribution from the cirrus for the tenuous plumes for PCC. The problem will be exacerbated for other volcanoes with relatively tenuous plumes, since we use the same threshold for all cases. We found using the color ratio threshold of 0.5 does a reasonably good job of retaining the ash and sulfate plumes from PCC and Nabro volcanoes.

**Specific comments about figures**

In a lot of the figures the axes and colorbar labels are missing. Also some of the labels are not written clearly. For example, the authors use underscores and abbreviations. I think using proper label names with appropriate variable symbol definitions and units would make the figures clearer. Also latitude/longitude units should use the degree symbol (not the abbreviated 'deg'). At the very least, the labelling should be consistent throughout the paper.

We have addressed these issues in the revised version.

**P1L28-30: There is also large disagreement (>100%) between CALIPSO and SAGE-III at altitudes below 20 km (Figure 11b). This should be stated in the abstract.**

We have added the following to the abstract:

"Similarly there are large differences ( $\geq 100\%$ ) within 2 to 3 kilometers above the tropopause which

might be due to cloud contamination issues."

**P3L26: I see two Kar et al. (2018)s in the references section. Please use 'a' and 'b' to differentiate between them.**

Done.

**P4L10-11: 'The consequences of this change...' - I suggest adding the V3 zonally and vertically averaged attenuated scattering ratio (for the same time period) to Fig. 1. This would make the change from V3 to V4 very clear.**

Per your suggestion we have now revised Figure 1. We are now showing the median values, as they better represent the distribution of CALIPSO data.

**P4L17: Change 'over this latitude' to 'over each latitude'.**

Done.

P5L17: 'accurate to about 1%' - do you have a reference for this?

It is actually 1.6% and we have corrected this in the revised version. This is from Kar et al. (2018a)

and we have added the reference here.

P5L28: Delete 'going'. P6L8: Replace 'i.e.' with 'such as'.

Done.

P6L11: 'Vaughan et al. (2009)' - Is there a new reference for the V4 level two layer detection algorithm that you could add?

The layer detection algorithm has not changed in version 4.

P6L14: Replace 'but' with 'however'. P6L18: Change 'product' to 'level 3 stratospheric aerosol product'. P6L21: Change 'the primary input files used for this product' to 'the primary input file used for the present product'.

Done.

P7, Figure 2: For consistency, should use small 'b' in the 'Write results to Background component' box.

Done.

**P7L7: Please define the 'local tropopause'. E.g. is this taken from GMAO?**

We have added the following:

"The tropopause heights were taken from the Modern-Era Retrospective analysis for Research and Applications 2 (MERRA-2) reanalyses as in all V4 products (Gelaro et al., 2017)."

**P8L4: Change 'Antarctica' to 'Antarctic'. Change 'both the hemispheres' to 'both hemispheres'.**

Done.

**P9L24: Shouldn't this be 'Vernier et al. 2013'?**

Actually it was first described in Vernier et al., 2009.

**P9L25: The Puyehue ash did not reach 17 km. Maximum heights observed by CALIOP were \_13 km (Vernier et al., 2013; Prata et al., 2017).**

This sentence has been deleted in the revised version and the text has been restructured.

**P9L26: Threshold of 0.5 seems too low. Vernier et al. (2013) use a threshold of 0.8 to discriminate between clouds and volcanic ash. I think the impact of this threshold should be mentioned (see specific comments above).**

We have already responded to this above.

**P10L11: Change 'threshold' to 'threshold of the level 2 layer detection algorithm'.**

Done.

**P11, Figure 4: Can you comment on what's causing the high scattering ratios just above 10 km at \_50\_N?**

The first referee has pointed out that this could be due to the Grímsvötn volcano and we have added the following:

"Further, the high scattering ratios near 50°N are likely due to the Grímsvötn volcano, which erupted in May 2011."

**P12L3: What does this look like for a threshold of 0.75-0.80? You may get more of a signal for the Puyehue event.**

As we mentioned above, taking a higher threshold also increases the cirrus contamination.

**P12L4: Change 'Nabro' to 'Nabro (near 30\_N)'.**

Done.

comment).

**P12, Figure 5: I think the labels are wrong here i.e. Figure 5b looks like 'background' and Figure 5a looks like 'all aerosol'. P12L19-21: I don't see a high number of samples over North America in Figure 5b (see above Figure 5**

Actually, the labels are correct, as can be verified from the attached color scale. In order to make it more clear and per the first referee's suggestion, we have now added an additional plot showing the difference in sample numbers between the two modes. We are now showing the sample number distribution at 17 km, showing enhanced sampling over the Asian summer monsoon plume area.

**P13L27: Change 'significant ash' to 'significant ash and sulfate'.**

Done.

**P14L22: Change 'image' to 'figure'. P14L23: Change '(Kasatochi, Nabro etc.) to '(e.g. Kasatochi and Nabro)'.**

The text has been revised here in view of two new panels and these words do not appear in the new text.

P15L10: 'quite clearly seen' - I'm not sure it is that clear. There are several other features in the figure that are more apparent than the Black Saturday bushfires, which aren't commented on. I suggest changing to 'can be identified'.

Done.

P15, Figure 6: I think you could add panels representing middle and high latitude bands to better represent the major stratospheric perturbations on a global scale (see specific comments above). Also, there's a significant feature around December 2010 that's not mentioned. This was probably due to the Merapi (7.54\_S, 110.446\_E) eruption in Indonesia in November 2010. Surono et al. (2012) estimate 0.44 Tg of SO2 in the upper troposphere and the plume reached heights of 16-17 km. Another feature that's not explained is the one around July 2015. It seems quite significant. Do you know what's causing it?

Per your suggestion we have now added two new plots (now Figure 7) showing the stratospheric features at mid/high latitudes (40°S-60°S and 40°N-60°N). Originally we were only trying to point out the most prominent cases without cluttering it too much with labels. However we have now pointed out Merapi as well as some other features in the plot. The feature near July 2015 is the

signature of Calbuco volcano (also shown now in Figure 7). The Calbuco signature at these latitudes could be seen for several months afterwards.

**P16, Figure 7d: Please fix the cropping at the bottom of the figure - some text has been cropped.**

Done.

**P16L15-16: 'irregular shapes' - could this also be due to ice particles?**

While the differential attenuation strongly suggests these are mostly smoke particles, there is always the possibility of ice formation from the pyroconvection event. This sentence has been modified as:

"The high volume depolarization ratio ( $\geq 0.1$ ) seen in Figure 7b is somewhat unusual for smoke and suggests the presence of irregular soot particles and mineral dust and possibly some ice particles, with fast adiabatic lifting possibly retaining the initial irregular shapes (Haarig et al., 2018, Khaykin et al., 2018)."

**P17L6: 'smoke spreads globally' - I don't see this in the figure. It looks like the smoke spreads throughout the Northern Hemisphere but the Southern Hemisphere scattering ratio remains unchanged.**

What we meant was that the plume spread at all longitudes and at lower latitudes---in any case "globally" has now been deleted and the new text reads as:

"After the original injection of smoke in August 2017 at mid-latitudes, the smoke spreads to lower latitudes as can be seen in these monthly mean spatial distributions from the level 3 stratospheric aerosol product."

**P17, Figure 8: What is the cause of the high scattering ratio from 25-30 km over the equator?**

The high scattering ratios at 25-30 km (now in Figure 9) reflect the tropical stratospheric reservoir. We have added the following:

"As in Figure 4, the feature with high attenuated scattering ratio near 25-30 km seen in all the four panels is the signature of the tropical reservoir of stratospheric aerosols, maintained by a complex interplay of transport from the troposphere and stratospheric dynamics as well as microphysical processes including the Brewer-Dobson circulation, the QBO, evaporation and sedimentation (Trepte and Hitchman, 1992, Kremser et al., 2016)."

**P17L17: Kelud erupted in February 2014 not April 2014 (see Kristiansen et al., 2015).**

Thank you—we have corrected this in the text.

P18L1: 'The gradual lofting of the plume from around 17 km over the tropics to nearly 24 km over several months...'. This seems to imply a rise of 7 km, which I think is misleading. Measuring from the top of the aerosol feature it looks like it rises from 21 to 24 km from March-December 2014 (a rise of 3 km). Please clarify this in the text.

We have now revised the sentence as:

"The gradual lofting of the plume, with its top rising from ~21 km over the tropics in March to ~24 km in the same general location several months later, shows the signature of stratospheric dynamics in the CALIPSO stratospheric aerosol product."

P19L7-9: The Calbuco volcanic cloud actually went almost directly through the SAA (see http://nicarnicaaviation.com/ calbuco-eruption-april-2015). Eventually it spread through the Southern Hemisphere but due to the rejection of data in the SAA region a large proportion of the Calbuco signal may not be captured in the CALIPSO level 3 stratospheric aerosol product. I think this is worth mentioning here.

We have revised the text as follows:

"The initial plumes would be missed out in the level 3 stratospheric aerosol product because data over the SAA region were not included. However the plumes quickly spread around the southern hemisphere in a belt between 60°S to 30°S (Lopes et al., 2019) and can be seen in the level 3 stratospheric aerosol product from May 2015 onwards for several months."

**P20L18: Change 'essentially same' to 'essentially the same'.**

Done.

**P20, Equation (4): I got slightly confused here with the notation. What's the difference between $\alpha_P(r)$ (defined at P13L24) and $\sigma(z)_{CALIPSO}$ ? And which variable is the one that corresponds to the 'all aerosol' profile product?**

Thanks for pointing out this oversight—we have replaced  $\alpha$  by  $\sigma$ , which would make it consistent everywhere. The same variable will represent the extinction coefficient in both the components and for comparisons with SAGE III we have used only the "all aerosol" component, since the latter includes extinctions from all sources.

**P22, Figure 11 caption: 'the mean 532 nm extinction coefficient' is this what $\sigma(z)_{CALIPSO}$ is? In Eq. (4) the definition is the 'extinction coefficient at altitude z'. I would use the same wording to avoid confusion or put the symbols ( $\sigma(z)_{CALIPSO}$ and $\sigma(z)_{SAGE}$ ) in parentheses in the figure caption.**

The extinction coefficient as a function of altitude is  $\sigma(z)$  with subscripts either CALIPSO or SAGE. The curves in Figure 11 (now Figure 12) represent the mean taken over all the profiles of  $\sigma(z)$  using all the data as mentioned in the legend. We think it is generally clear.

**P22L18: 'the presence of clouds which may impact the retrievals' - Please provide a little more information on how clouds impact the retrieval. If some clouds weren't removed, wouldn't this bias SAGE-III aerosol extinction high? Thus compensating for the difference seen in the comparison with CALIPSO below 20 km?**

In the revised version, we have discussed the possible impacts of cloud clearance issues in our product in section 2.2.1 and added a new figure (Figure 4). Further we have added the following "As pointed out in section 2.2.1, the filtering scheme that removes thin cirrus clouds in the all aerosol mode is not as efficient as the technique employed in the background mode. Consequently, scattering artifacts from undetected subvisible cirrus are more likely to appear in the all aerosol mode in the tropical lower stratosphere within a few kilometers above the tropopause. Using the

extinction profiles from the background mode reduces the differences at these altitudes but does not completely eliminate them (not shown)."

P22L23: Change '2.0' to '2'.

Done.

P22L24: On my first read through, I immediately thought the assumption of constant lidar ratio was the issue. You go on to discuss this but it's not mentioned here. Perhaps it's worth adding a sentence and referencing the discussion that comes later.

We have added the following:

"We further discuss the possible issues resulting from uncertainties in lidar ratios below."

P23L15: Change 'Discussion:' to 'Discussion'.

Done.

P24L3-5: Change 'tropical latitudes' to 'tropical latitudes (30\_S–30\_N)'.

Done.

P24L5: Change 'higher latitudes' to 'higher latitudes and lower altitudes'.

Done.

**P24L10: 'smoke, marine aerosols etc' - please list all the aerosol types considered by references cited.**

All the different types of aerosols in the troposphere are really not relevant in the stratosphere, so we have simply deleted "such as smoke, marine aerosols etc."

**P24, Eq. (5): In Eq. (3), the two-way particulate transmittance is range-dependent. I assume it would be range-dependent ( $T_{2p}(r)$ ) here too?**

Thank you and this has been corrected.

P25L16: 'substantially lower' - Could you put a number to this? E.g. what's the mean lidar ratio in the lowermost stratosphere? I think it is important to give a number or range given the discussion that follows.

We have added " $\leq 40$  sr" to incorporate both the lowermost stratosphere as well as the high latitudes.

P27L3: 'no significant injections of ash from volcanoes' - this is probably true, but there were significant injections of SO2 and therefore sulfate. For example, Ambae (Vanuatu) in April 2018 underwent a significant SO2-rich eruption (see specific comments above).

We have now revised Figure 13 (now Figure 14) by removing possibly contaminated data from the pyroCb event of 2017 and all data after March 2018 to avoid effects from Ambae volcano.

P27L12: Change 'volcanic eruptions' to 'volcanic eruptions and wildfires'.

Done.

P27L16: Change 'mid-to-high latitudes' to 'mid-to-high latitudes (30\_S–60\_S and 30\_N–60\_N)'

Done.

P27L16: Change 'high altitudes' to 'high altitudes (10–20 km)'

Done.

---

## Author Comment (AC4)

**Response to Referee#4**

The paper "CALIPSO Level 3 Stratospheric Aerosol Product: Version 1.00 Algorithm Description and Initial Assessment" presents and discusses the new science algorithm and data handling techniques that are developed to generate the CALIPSO version 1.00 level 3 stratospheric aerosol profile product. The study falls within the scope of AMT. The authors have done a thorough job, the manuscript is wellwritten/structured, the presentation clear, the language fluent, the quality of the figures high. The result support the conclusions.

Thanks very much for carefully reading the manuscript and for your useful suggestions.

Two major deficiencies are the implementation of a constant stratospheric aerosol lidar ratio (50 sr), regardless of an aerosol type classification, and the evaluation of the stratospheric aerosol product against SAGEIII extinction coefficient observations, a product which has not been validated (including issues of SAGEIII such as cloud contamination propagating in the comparison). However the stratospheric aerosol product and all the issues are properly and extensively discussed in the manuscript, thus I recommend publication in AMT, under minor revisions before it can proceed to be published.

For this first version of the product we have used a constant lidar ratio of 50 sr globally. In future versions we hope to incorporate a more informed lidar ratio distribution. As shown in Figure 14, by making use of the extinction retrievals from SAGE III and backscatter measurements from CALIPSO, an initial estimate of the stratospheric aerosol lidar comes out to be  $\sim$ 46±6 sr, quite close to the value we have adopted. Although not validated yet, the SAGE III extinction retrievals are still the gold standard for stratospheric aerosol data and so we have used them in our initial comparisons.

**Minor comments:**

1) P1L17-18: "gridded level 3 product is based on version 4.2 of the CALIOP level 1 and level 2 data products". According to this sentence CALIOP level 1 V4.2 is used. It is not clear whether the authors refer to Level 1B or Level 1.5 Profile Data. In the case of L1B, please provide a web link to the used data repository.

Added 1B in P1 and given the link to the data repository in section 2.1.

2) P1L27: "where the average difference between zonal mean extinction profiles is typically less than 25% between 20km and 30km". Please rephrase to provide also whether the sentence refers to overestimation or underestimation compared to SAGEIII.

We have added "(CALIPSO biased high)" within brackets.

3) P3L29-30: "This is a level 3 monthly averaged product gridded in latitude (50), longitude (200) and altitude (900m)". Although the justification of the 900m vertical resolution is sufficient, the authors should provide explanation on the reasons why the spatial resolution of 50x200 deg2 grids was selected. How much the selected spatial (horizontal), vertical and temporal resolution affect the final dataset (in terms of backscatter and extinction coefficient profiles at 532nm)?

We have added the following in the text:

"Given the low SNR in the stratospheric backscatter measurements, it is necessary to average the data substantially, both spatially and temporally. Averaging the backscatter data over 5° in latitude increases the SNR by a factor of 40 (compared to single shot profiles) and provides a reasonable

depiction of stratospheric aerosol distribution. This is also consistent with the early results of Thomason et al. (2007), who used the early CALIPSO measurements together with data from the CALIPSO simulator (Powell, 2005) to show that averaging the data over  $5^{\circ}$  in latitude and about 1 km in the vertical resulted in fairly representative stratospheric distribution. Further, spatial distributions of stratospheric species tend to be zonally symmetric (e.g. Kremser et al., 2016). In order to capture the signature of any possible longitudinal variation, e.g., the Asian Tropopause Aerosol Layer (ATAL) which occurs over Asia every summer during the monsoon months, we have used a longitudinal grid of  $20^{\circ}$ ."

It is important to remember that we are trying to retrieve very low extinction coefficients  $(\sim 10^{-4} - 10^{-5} \text{ km}^{-1})$  in the stratosphere using the backscatter measurements with low SNR at those altitudes. This cannot be achieved without a significant amount of averaging of the data spatially and temporally. It may be mentioned that the currently available Global Space-based Stratospheric Aerosol Climatology (GLOSSAC) product that uses CALIPSO data as well as OSIRIS data is zonally invariant and binned in 5 degree in latitude and produced on monthly basis, consistent with the needs of the climate modelling community.

We believe the justification for the adopted grid lies in capturing the various stratospheric perturbations in the retrieved products, to the extent possible. As shown in the manuscript, seasonal and regional perturbations like the Asian Tropopause Aerosol Layer (ATAL) as well as the pyroCb events and volcanic signatures are well captured in the product, vindicating the spatial and temporal resolution adopted for the product.

**4) P5L26: "Note that the range of altitudes to be covered in the stratosphere at various latitudes is from 8.2 km to 36 km, the latter being the lower limit of the calibration region". Please mention the applied methodology of decoupling stratospheric and tropospheric layers, since the altitude of 8.2 km frequently lies below Tropopause? Does it rely on MERRA-2 by GMAO?**

We have added the following:

"The altitude resolution of the CALIOP level 1 profiles varies with altitude from 60 m between 8.3 km and 20.2 km to 180 m between 20.2 km and 30.1 km and finally to 300 m between 30.1 km and 40.0 km. In order to achieve a uniform altitude resolution, the vertical grid resolution was set to 900 m. Note that the tropopause can occur below 8.3 km at high latitudes, but the vertical resolution of level 1 profiles changes again below this altitude and the lower limit was kept at 8.3 km as a trade-off between computational complexity and the stratospheric information content, while the upper limit was set at 36 km, which is the lower limit of the calibration region. The tropopause heights were taken from the Modern-Era Retrospective analysis for Research and Applications 2 (MERRA-2) reanalyses as in all V4 products (Gelaro et al., 2017)."

5) P8L8-9: "Further, all L1B profiles within the South Atlantic Anomaly (SAA) region are also removed": Why do the authors remove CALIOP observations over the SAA region. Based on Kar et al., 2017 (CALIPSO lidar calibration at 532âA° L'nm: version 4 nighttime algorithm), the new nighttime CALIOP calibration technique compensates for the higher NSR values, resulting in reliable calibration

**coefficients even over the SAA region. The authors it is suggested to include the justification in the manuscript.**

While the version 4 calibrations were done over SAA region also, the stratospheric profiles over this area may still be noisy. Therefore, for this first version of the product, we decided to remove all data over the SAA region. In fact this issue has been discussed in detail in the second paragraph on this page.

**6) P8L25: "... leading to generally lower CAD scores (Liu et al., 252019).". Since CAD ranges between - 100 and 100, it is not clear whether the authors refer to more aerosol reliable retrievals (CAD -> -100) or to absolute values of CAD score, therefore, CAD values closer to zero.**

We have revised the text as "generally lower absolute values of CAD scores".

**7) P9L4-7: Although the authors provide the Vaughan et al., 2009 reference, some information on the noise filter should also be included in the manuscript, even if briefly.**

We have modified the text adding some information on the noise filters as:

"Essentially a range dependent threshold array of attenuated scattering ratios is constructed, which incorporates noise from two types of sources. The first category is the range invariant noise and includes detector dark noise and noise from the solar background light. The second category is the range dependent noise from single shot measurements and is calculated from the molecular models. Using this range dependent threshold, outliers are removed (for details see Vaughan et al., 2009, section 2c). After removing the outliers, the 5 km profile is assigned to the appropriate spatial grid."

**8) Figure 4: Based on the manuscript, Figure 4b and 4c refer to the aerosol mode, however it is not clear neither in the caption nor in the manuscript whether they refer to the background or the aerosol mode. In addition, high stratospheric values are observed at 0o latitude, between 25 and 30 km height. Where do the authors attribute the observed values?**

Figure 4 (now Figure 3 in the revised version) is showing the effects of the various filters. Figure 3a is the standard background mode where a depolarization filter is employed after removing all layers or features. Figure 3b is not really part of the product but is only meant to show the result, if we had employed a depolarization filter after retaining the aerosol layers in the stratosphere— clearly the Cordon plume is being missed here. Figure 3c is the standard "all aerosol" mode" where a color ratio filter is used after retaining the aerosol layers and shows that this filter captures both the volcanoes. We have modified the caption as:

"Zonally averaged height-latitude cross sections of attenuated scattering ratio for June 2011: a) after removing all detected layers and using a volume depolarization ratio filter (i.e., background aerosol only); b) including aerosol layers in the stratosphere detected by the level 2 algorithms with a 5% volume depolarization ratio filter applied; and c) including the level 2 aerosol layers but using an attenuated color ratio filter instead of the volume depolarization ratio filter. The white area in the northern high latitudes in summer indicates lack of nighttime data."

As regards the high values over the tropics we have added the following sentence:

"Note that the enhanced scattering ratios near 25-30 km represent the tropical reservoir of stratospheric aerosols (Trepte and Hitchman, 1992, Kremser et al., 2016)."

**9) P12L5: "Note the high scattering ratio values in the Antarctic latitudes between 15 km and 25 km". The authors are kindly requested to provide a reference for this statement.**

No reference is available for this statement---essentially this is what we are pointing out from CALIOP measurements.

**10) P12L17-18: "The white grid cells over southeast Asia occur because the tropopause is higher than 16 km in this region". The authors are kindly requested to provide a reference for this statement, including the typical tropopause height over this region.**

Note that this Figure has now been revised and we are now showing the sample number distribution at 17 km, where no data drop out can be seen from tropopause-related issues.

**11) P12L21-22: "This is again likely due to small particles which are in the process forming PSCs". The authors are kindly requested to provide a reference for this statement.**

This particular sentence has now been deleted and the sentence preceding this has been modified as:

"Also note the high number of samples over parts of Antarctica, partly due to oversampling from orbital configuration and related to small particles below the detectability of PSCs by the PSC mask algorithm."

**12) P13L13: "For the CALIPSO stratospheric aerosol product, the particulate multiple scattering factor is taken as 1 for all species of stratospheric aerosols". The authors are kindly requested to provide a reference for this statement. Which is quantitative the effect of this assumption on the discussed stratospheric aerosol product?**

We expect the single scattering assumption to hold in the stratosphere most of the time, except may be for the early part of the plume injection. This is also consistent with the retrievals in the version 4 level 2 aerosol retrievals

We have revised the relevant text as:

"For the CALIPSO stratospheric aerosol product, the particulate multiple scattering factor is taken as 1 for all species of stratospheric aerosols, consistent with the approach taken in the CALIPSO level 2 aerosol retrievals (Young et al., 2013; Young et al., 2016; Young et al., 2018)."

A quantitative discussion of this is examined in the comprehensive error analysis detailed in Young et al., (2013).

13) P13 - Stratospheric Aerosol Lidar Ratio of 50 sr is used. Although the authors explain in detail the selection of the specific lidar ratio value and evaluate against SAGEIII observations, it is expected that the uniform value used globally, regardless of the aerosol type, introduces large uncertainties. Which is the effect of this assumption to the stratospheric aerosol product? The authors mention that appropriate LR values for different aerosol subtypes will be introduced in future versions of the stratospheric product, however the assumption of constant LR value highly affects the reliability of the extinction coefficient profiles and should be mentioned in the abstract.

We believe we have adequately discussed the lidar ratio issue in the discussion section. In particular the lidar ratio obtained by using the extinction from SAGE III and backscatter from CALIOP gives a value quite close to the adopted lidar ratio in much of the stratosphere.

We have revised the relevant sentence in abstract as:

"Further, we show that the extinction profiles (retrieved using a constant lidar ratio of 50 sr) capture the major stratospheric perturbations in both hemispheres over the last decade resulting from volcanic eruptions, extreme smoke events, and signatures of stratospheric dynamics."

**14) Figure 8: The authors should discuss on the high values of attenuated scattering ratios observed over the equator, including the proper references.**

We have added the following:

"As in Figure 3, the feature with high attenuated scattering ratio near 25-30 km seen in all the four panels is the signature of the tropical reservoir of stratospheric aerosols, maintained by a complex interplay of transport from the troposphere and stratospheric dynamics as well as microphysical processes including the Brewer-Dobson circulation, the QBO, evaporation and sedimentation (Trepte and Hitchman, 1992, Kremser et al., 2016)."

**15) P18L1-5: "The persistence of the stratospheric perturbation for several months is consistent with the results of Vernier et al. (2016) who found the presence of ash in the lower stratosphere 3 months after the Kelud eruption from balloon observations". The observed features are qualitative consistent with the results of Verner et al. (2016). Is it possible to the authors to include a quantitative comparison?**

It is not possible to do a comparative comparison, because we are using a monthly averaged product at 5° latitude by 20° longitude—for a proper profile by profile comparison we need to have proper collocations. In any case, this paper is primarily devoted to a description of the algorithm and data product. While we offer an initial assessment of data product quality by providing multiple comparisons to SAGE III measurements, this paper is not intended as a comprehensive validation of all possible results that could be retrieved from this product.

16) P20L3-5: "SAGE III performs solar and lunar occultation measurements as the ISS orbits the Earth and covers the entire global latitude (90oS to 90oN) and longitude range (180oW to 180oE)." ISS orbital characteristics are characterized by 51.60 inclination, therefore the authors it is suggested to check the global latitude coverage (90oS to 90oN).

We have modified this sentence to:

"SAGE III performs solar and lunar occultation measurements as the ISS orbits the Earth and covers a broad latitude band (60°S to 60°N) and longitude range (180°W to 180°E)."

17) P20L15-17: "The globally averaged value of the Angstrom exponent derived using all 15 months of data is about 1.56". Please mention between which wavelengths.

In the preceding sentence we had already mentioned that the Angstrom exponent was derived from 521 nm and 1022 nm. In any case we have added "(between 521 nm and 1022 nm)" once again.

**18) P20L22: " $\sigma(z) = 100 \times (\sigma(z)CALIPSO - \sigma(z)SAGE)/\sigma(z)SAGE$ ". How are extreme cases treated? Which computational filters are applied? For instance, cases with $\sigma(z)CALIPSO = 0$ ( $\sigma(z) = -1$ ), or cases with very low values of $\sigma(z)SAGE$ are also included? In case of applied filters in the dataset used prior to the results, the authors should mention them in the manuscript.**

As clearly mentioned in the text, we have used the extinction coefficients with fractional uncertainties less than 100% as retrieved from both SAGE III and CALIPSO. No other filters were used for these comparisons.

**19) P22L14: "between CALIPSO and SAGE III extinction at all altitudes with CALIPSO having a high bias". Wherever the manuscript refers to statistical indicators, such as the "high bias" here, the authors should mention the corresponding computed values.**

Done.

**20) P23L8: "calculated using the average extinction coefficient profiles between 20 km and 30 km". The reason of selecting vertically the region between 20km and 30km and not the region from 20 km up to 34 km, hence including the stratospheric region of V3 calibration, is not clear nor justified in the manuscript, since it is proven in Kar et al. (2017) that this region is not aerosol free.**

The retrieved profiles above 30 km are often quite noisy from both the instruments and were not used. Once again, retrieving aerosol extinction coefficients  $\sim 10^{-5}$  km-1 from these measurements is stretching the limits of the instruments' measurement capabilities while at the same time not adding anything significant to the stratospheric optical depth estimates. The purpose of this figure is to show the difference in the retrieved optical depths within the altitudes where much of the stratospheric aerosol generally resides.

**21) P2314: "though the differences begin to rise substantially in the midlatitudes of both hemispheres". Please include explanation on the observed features, including the necessary references.**

We don't completely understand the cause of these differences at this time. Further validation work using data from other instruments providing extinction retrievals might be of help in determining these.

**References:**

- Thomason, L.W. et al., Atmos. Chem. Phys., 7, 5283-5290, https://doi.org/10.5194/acp-7-5283-2007, 2007.
- Powell, K. A., Development of the CALIPSO lidar simulator, M. S. thesis, College of William and Mary, 228 pp. available at http://www-calipso.larc.nasa.gov/resources/publications. php

---

## Author Response (AR1)

**CALIPSO Level 3 Stratospheric Aerosol **Profile** Product: Version 1.00 Algorithm Description and Initial Assessment**

5 Jayanta Kar1,2, Kam-Pui Lee1,2, Mark A. Vaughan2, Jason L. Tackett1,2Tackett2, Charles R. Trepte2, David M. Winker2, Patricia L. Lucker1,2, Brian J. Getzewich1,2

[revised manuscript text omitted]

As shown in Eq. (1), the attenuated scattering ratios, R'(z), are computed as the ratio of the measured attenuated backscatter coefficients,  $\beta'_{measured}(z)$ , which contain contributions from both

molecular and particulate backscatter ( $\beta_m(z)$  and  $\beta_p(z)$ , respectively), and the attenuated backscatter coefficients calculated from modeled profiles of molecular number densities,  $\beta'_{modeled}(z)$  (Vaughan et al., 2009).

$$R'(z) = \frac{\beta'_{measured}(z)}{\beta'_{modeled}(z)} = \frac{\left(\beta_{m}(z) + \beta_{p}(z)\right) T_{m}^{2}(z) T_{O_{3}}^{2}(z) T_{p}^{2}(z)}{\beta_{m,modeled}(z) T_{m,modeled}^{2}(z) T_{O_{3},modeled}^{2}(z)} = \left(1 + \frac{\beta_{p}(z)}{\beta_{m,modeled}(z)}\right) T_{p}^{2}(z)$$
(1)

In this expression,  $T_x^2(z)$  represents the two-way transmittance (i.e., signal attenuation) between 5 the lidar and altitude z for air molecules (X = m), ozone  $(X = O_3)$ , and particulates (X = p). In version 3 (V3), the calibration region was fixed at 30-34 km, with the assumption that the aerosol loading in this region was negligible (Powell et al., 2009); i.e.,  $\beta_p(z) \approx 0$  and  $T_p^2(z) = 1$ . This assumption essentially forces forced the V3 attenuated scattering ratios in the region to one. For the V4 data release, the calibration region was raised to 36-39 km, with the concomitant 10 assumption that the mean scattering ratio at these higher altitudes is  $1.01 \pm 0.01$ . The V4-data attenuated scattering ratios now (correctly) show significant aerosol in the altitude region used for the V3 calibration, with a strong maximum appearing over the tropics (Figure 1). The V4 data also capture the seasonal variation of these scattering ratios (Kar et al., 20182018a, see their Figure 12). This improved calibration in V4, now accurate to about 1.6%, provides the motivation for the 15 development of the CALIPSO stratospheric product, as it enables the retrieval of aerosol extinction coefficients in regions previously (but incorrectly) assumed to be aerosol-free- (Kar et al., 2018a).

**2.2 Design and algorithm description**

The level 3 stratospheric aerosol profile product reports height-resolved monthly mean profiles of aerosol backscatter and extinction coefficients on a uniform spatial grid that extends 5° in latitude (from 85°N to 85°S), 20° in longitude (from 180°W to 180°E), and 900 m in altitude. This grid was chosen to allow for adequate averaging of the data to compensate for the lowerGiven the low SNR in the stratosphere while retaining some level of zonal information. Note that the range of altitudes to be covered in the stratosphere at various latitudesstratospheric backscatter 25 measurements, it is necessary to average the data substantially, both spatially and temporally. Averaging the backscatter data over 5° in latitude increases the SNR by a factor of 40 (compared to single shot profiles) and provides a reasonable depiction of stratospheric aerosol distribution. This is also consistent with the early results of Thomason et al. (2007), who used the early

CALIPSO measurements together with data from 8.2-the CALIPSO simulator (Powell, 2005) to show that averaging the data over  $5^{\circ}$  in latitude and about 1 km to 36 km, the latter being the lower limit of the calibration region. in the vertical resulted in fairly representative stratospheric distribution. Further, spatial distributions of stratospheric species tend to be zonally symmetric

(e.g. Kremser et al., 2016). In order to capture the signature of any possible longitudinal variation, 5 e.g., the Asian Tropopause Aerosol Layer (ATAL) which occurs over Asia every summer during the monsoon months, we have used a longitudinal grid of 20°. The altitude resolution of the CALIOP level 1 profiles varies over this with altitude range, going from 60 m between 8.23 km and 20.2 km to 180 m between 20.2 km and 30.1 km and finally to 300 m between 30.1 km and 40.0 km. In order to achieve a uniform altitude resolution, the vertical grid resolution was set to 10 900 m. Note that the tropopause can occur below 8.3 km at high latitudes, but the vertical resolution of level 1 profiles changes again below this altitude and the lower limit was kept at 8.3 km as a trade-off between computational complexity and the stratospheric information content, while the upper limit was set at 36 km, which is the lower limit of the calibration region. The tropopause heights were taken from the Modern-Era Retrospective analysis for Research and Applications 2 15 (MERRA-2) reanalyses as in all V4 products (Gelaro et al., 2017). In the current version of the stratospheric aerosol product we use only nighttime data as they have significantly better SNR as

compared to the daytime data (Hunt et al., 2009).

Each level 3 stratospheric aerosol file reports two distinct realizations of the monthly 20 averaged data products. The first of these is the "background" mode, which is designed to represent the long-term background stratospheric aerosol loading. In order to achieve this, we need to remove all readily detectable perturbations within the stratosphere; i.e., such as overshooting cirrus clouds, polar stratospheric clouds (PSCs), and strongly scattering injections of smoke, volcanic ash, and other aerosol species which are detected using the layer detection algorithm implemented in the CALIOP level 2 data processing (Vaughan et al., 2009). The second 25 realization is the "all aerosols" mode which is designed to represent the time history of aerosol loading in the stratosphere resulting from all possible sources. In this case, the cirrus clouds and PSCs are still removed, exactly as is done for the background mode, buthowever, subject to various quality assurance tests, the aerosol layers detected in the level 2 analyses are retained. Details of the averaging algorithms and the various data filtering schemes are provided in the following 30 sections.

**2.2.1 Gridding and filtering**

5

10

The overall design of the level 3 stratospheric aerosol product is shown in Figure 2. To begin with, three input files are required for each granule under consideration. A CALIOP granule comprises half an orbit of data either from the daytime or the nighttime part of the orbit and divided by the day-night terminator. As noted in section 1, the primary input files used for this present product is are the lidar level 1B file, with the corresponding level 2 5 km merged layer and PSC mask files (Pitts et al., 2009) used for filtering. While the level 1B and level 2 merged layer files are based on V4, the currently available level 2 PSC files are based on V3. The latter is only available as a daily file and not for each granule separately. The 5 km merged layer file is a new product in V4 that reports the locations of all aerosol and cloud layers detected at both 5 km (also 20 km and 80 km) and single shot (333m) resolution (Vaughan et al., 2016).